Clinical Science and Epidemiology

# Genomic and Resistance Epidemiology of Gram-Negative Bacteria in Africa: a Systematic Review and Phylogenomic Analyses from a One Health Perspective

Ⓘ John Osei Sekyere,[a] Melese Abate Reta[a]

[a]Department of Medical Microbiology, School of Medicine, Faculty of Health Sciences, University of Pretoria, Pretoria, South Africa

**ABSTRACT** Antibiotic resistance (AR) remains a major threat to public and animal health globally. However, AR ramifications in developing countries are worsened by limited molecular diagnostics, expensive therapeutics, inadequate numbers of skilled clinicians and scientists, and unsanitary environments. The epidemiology of Gram-negative bacteria, their AR genes, and geographical distribution in Africa are described here. Data were extracted and analyzed from English-language articles published between 2015 and December 2019. The genomes and AR genes of the various species, obtained from the Pathosystems Resource Integration Center (PATRIC) and NCBI were analyzed phylogenetically using Randomized Axelerated Maximum Likelihood (RAxML) and annotated with Figtree. The geographic location of resistant clones/clades was mapped manually. Thirty species from 31 countries and 24 genera from 41 countries were analyzed from 146 articles and 3,028 genomes, respectively. Genes mediating resistance to $\beta$-lactams (including $bla_{TEM-1}$, $bla_{CTX-M}$, $bla_{NDM}$, $bla_{IMP}$, $bla_{VIM}$, and $bla_{OXA-48/181}$), fluoroquinolones ($oqxAB$, $qnrA/B/D/S$, $gyrA/B$, and $parCE$ mutations, etc.), aminoglycosides (including $armA$ and $rmtC/F$), sulfonamides ($sul1/2/3$), trimethoprim ($dfrA$), tetracycline [$tet$(A/B/C/D/G/O/M/39)], colistin ($mcr-1$), phenicols ($catA/B$, $cmlA$), and fosfomycin ($fosA$) were mostly found in *Enterobacter* spp. and *Klebsiella pneumoniae*, and also in *Serratia marcescens*, *Escherichia coli*, *Salmonella enterica*, *Pseudomonas*, *Acinetobacter baumannii*, etc., on mostly IncF-type, IncX$_{3/4}$, ColRNAI, and IncR plasmids, within *Intl*1 gene cassettes, insertion sequences, and transposons. Clonal and multiclonal outbreaks and dissemination of resistance genes across species and countries and between humans, animals, plants, and the environment were observed; *Escherichia coli* ST103, *K. pneumoniae* ST101, *S. enterica* ST1/2, and *Vibrio cholerae* ST69/515 were common strains. Most pathogens were of human origin, and zoonotic transmissions were relatively limited.

**IMPORTANCE** Antibiotic resistance (AR) is one of the major public health threats and challenges to effective containment and treatment of infectious bacterial diseases worldwide. Here, we used different methods to map out the geographical hot spots, sources, and evolutionary epidemiology of AR. *Escherichia coli*, *Klebsiella pneumoniae*, *Salmonella enterica*, *Acinetobacter baumannii*, *Pseudomonas aeruginosa*, *Enterobacter* spp., *Neisseria meningitis/gonorrhoeae*, *Vibrio cholerae*, *Campylobacter jejuni*, etc., were common pathogens shuttling AR genes in Africa. Transmission of the same clones/strains across countries and between animals, humans, plants, and the environment was observed. We recommend *Enterobacter* spp. or *K. pneumoniae* as better sentinel species for AR surveillance.

**KEYWORDS** One Health, antibiotic resistance, molecular epidemiology, diagnostics, genomics, resistome, mobilome, Africa

Address correspondence to John Osei Sekyere, jod14139@yahoo.com.

Antibiotic resistance is an understudied public health threat in Africa. Here, we provide the first comprehensive analysis of AR gene epidemiology and phylogenomics in Gram-negative pathogens from a One Health perspective.

Antibiotic resistance (AR), particularly in Gram-negative bacteria (GNB), is complicating infection management in Africa and the rest of the world as it restricts effective therapeutic options available to clinicians in human and veterinary medicine (1–3). Given the unsanitary environments common in developing countries as well as the limited health care and laboratory facilities, poor sewage management, high patient-to-physician ratios, little or no regulation on antibiotic usage, etc., the escalation of AR in developing countries (Africa) is precarious (3–5). Furthermore, there is limited funding available for molecular surveillance of AR to map out the true burden of the problem (2, 4). A recent global metagenomic survey, for instance, found that African countries had the highest AR gene (ARG) abundance, although in animals, Asia was found to have more AR hot spots than Africa (2, 5).

Together, GNB cause or aggravate some of the most fatal and common infections and diseases known to humankind: sepsis, meningitis/meningococcemia, gonorrhea, pneumonia, cystic fibrosis, urethritis, pelvic inflammatory disease, cholera, typhoid, whooping cough, diarrhea, etc. (6–15). It is worth noting that compared to Gram-positive bacteria, GNB have been implicated in the development of more resistance mechanisms, including resistance to last-resort antibiotics such as colistin (*mcr*), carbapenems ($bla_{NDM}$, $bla_{VIM}$, $bla_{KPC}$, $bla_{IMP}$, $bla_{GES}$, etc.), and tigecycline [*tet*(X)] (1, 16, 17). Moreover, these ARGs are found in more diverse GNB species and genera, confer multidrug resistance, and are associated with substantial morbidities and mortality (16–19). Hence, GNB species such as *Pseudomonas aeruginosa*, *Acinetobacter baumannii*, and *Enterobacteriaceae* expressing resistance to carbapenems and extended-spectrum $\beta$-lactamases (ESBLs) are classified as critical and high-priority pathogens, respectively, by the WHO (20, 21).

The presence of these ARGs and species in human, environmental, and animal samples in Africa (and globally) is well documented, strengthening the call for further One Health molecular surveillance, i.e., the periodic microbiological assessment and monitoring of AR in animals, humans, and environmental samples (2, 3, 5, 22). The importance of this One Health concept lies in the sources of AR. When AR-containing bacteria from farms end up in the environment through effluents and manure application on soils, they can be transferred to humans through vegetable and animal diets. As well, human misuse of antibiotics leads to the selection of AR, which is transferred into the environment and farms through hospital effluents, sewage treatment plants, etc. (23, 24) Yet, a systematic review on GNB in Africa from a One Health perspective is lacking (22, 25), limiting the comprehensive appreciation of the epidemiology of ARGs across the continent. Here, the epidemiology of GNB in Africa, their resistance genes, associated mobile genetic elements (MGEs), their evolutionary relationships, and geographical dissemination are described and analyzed. The sources of GNB, *viz.*, animals, humans, and the environment, are described to show the presence of GNB clones across humans, animals, and the environment using phylogenomics. This provides a novel insight in the evolutionary relationship between GNB species and clones, their geographical hot spots, and resistance determinants.

## RESULTS

**Characteristics of included studies.** The literature search returned 1,495 research articles: 1,490 articles (from PubMed, Web of Science, and ScienceDirect) and five articles from the references of the included articles. Duplicates were removed, and the remaining 868 nonduplicated articles' titles and abstracts were screened using the inclusion and exclusion criteria; this resulted in 309 articles. Full-text review of the remaining 309 manuscripts resulted in 146 studies being used for the qualitative and quantitative analyses after applying the inclusion and exclusion criteria (26) (Fig. 1A) (see also Tables S1 to S3 in the supplemental material).

The included articles spanned 31 countries from animal (A), human (H), and environmental (E) samples, with South Africa having the most (*n* = 24) studies and human samples being the most common source of the bacterial isolates. Samples

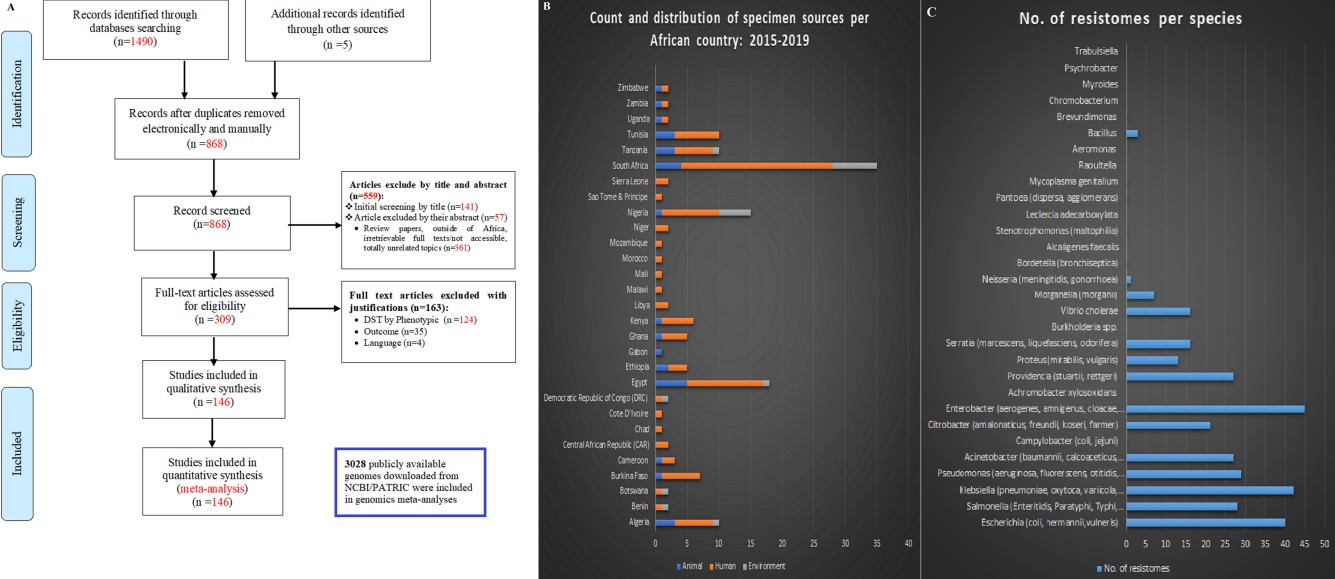

**FIG 1** Literature search strategy, inclusion and exclusion criteria, distribution of sample sources, and ARG count per species in Africa. (A) A systematic flow diagram summarizing the literature search methodology, databases used, and the inclusion and exclusion criteria adopted in getting the final 146 manuscripts used in these qualitative and quantitative analyses. Besides the included literature, 3,028 publicly available genomes were downloaded from PATRIC (https://www.patricbrc.org/)/NCBI's GenBank and analyzed to determine their resistomes and evolutionary geographic location. (Adapted from *PLoS Med* [70].) (B) Count of sample sources, *viz.*, animals, humans, and the environment, and geographical distribution of samples per country in Africa. (C) Counts of ARGs per species are depicted, showing that *Enterobacter*, *Klebsiella*, and *Escherichia* have the largest ARG repertoires of all the species (Table S7).

from animal sources (including animal food products) were also substantial while those from the environment were relatively few, and not all countries reported these samples (26) (Fig. 1B). These studies involved 23,157 isolates from 65,119 samples (isolation rate of 35.56%): 2,560 isolates from 5,950 animal (and animal foods) samples (43.03% isolation rate), 16,225 isolates from 57,464 human samples (isolation rate of 28.24%), and 4,372 isolates from 1,705 environmental samples (isolation rate of 256.42%). The various species identified in each study are summarized under the respective sample source in Table 1; the per-country breakdown is shown in Tables S1 to S3. Table S1 comprises GNB isolated from animal sources, Table S2 consists of GNB isolated from human sources, and Table S3 represents GNB isolated from environmental sources. Notably, the included articles did not undertake a One Health research on their own but focused only on clinical, animal, or environmental samples.

The 3,028 genomes (from African countries) included in this study were also obtained from animals, humans, plants, and the environment from 41 African countries: Angola, Benin, Botswana, Burkina Faso (B. Faso), Cameroon, Central African Republic (CAR), Chad, Comoros, Democratic Republic of Congo (DRC), Djibouti, Egypt, Eritrea, Ethiopia, Gambia, Ghana, Guinea, Guinea-Bissau, Kenya, Lesotho, Madagascar, Malawi, Mali, Mauritania, Mauritius, Morocco, Mozambique, Namibia, Niger, Nigeria, Republic of the Congo, Rwanda, Senegal, Sierra Leone, South Africa (S. Africa), Sudan, Tanzania, Togo, Tunisia, Uganda, Zambia, and Zimbabwe. Tables S4 to S6 contain information on the raw genomic metadata per species, color-coded AR gene data for each species, and phylogenomically ordered AR gene metadata per species, respectively.

**Species distribution (from included articles).** Of the 30 species isolated from the various human, animal, and environmental samples included in the studies used for this meta-analysis, the most common were *Escherichia* spp. (*n* = 9,292), *Klebsiella* spp. (*n* = 2,776), *Salmonella enterica* (*n* = 1,773), *Pseudomonas* spp. (*n* = 1,498), and *Acinetobacter* spp. (*n* = 705), which were all more often isolated from human samples than from animal or environmental samples; these statistics were also largely reflected in the species distribution in the genomics data (Table 1). These pathogens, including *Neis-*

**TABLE 1** Species distribution frequencies per sample source[a]

| Genus (species or serovar) | No. of animal samples (n = 5,950) | No. of human samples (n = 57,464) | No. of environmental samples (n = 1,705) | Total no. | No. of genomes included in phylogenomics |
|---|---|---|---|---|---|
| Escherichia (coli, hermannii, vulneris) | 1,566 | 4,899 | 2,827 | 9,292 | 592 |
| Salmonella (Enteritidis, Paratyphi, Typhi, Typhimurium) | 500 | 1,262 | 11 | 1,773 | 487 |
| Klebsiella (pneumoniae, oxytoca, variicola, michiganensis) | 204 | 2,528 | 44 | 2,776 | 311 |
| Pseudomonas (aeruginosa, fluorescens, otitidis, putida) | 45 | 1,366 | 87 | 1,498 | 95 |
| Acinetobacter (baumannii, calcoaceticus, haemolyticus) | 2 | 611 | 92 | 705 | 21 |
| Campylobacter (coli, jejuni) | 210 | 175 | 105 | 490 | 13 |
| Citrobacter (amalonaticus, freundii, koseri, farmeri) | 1 | 133 | 6 | 140 | 192 |
| Enterobacter (aerogenes, amnigenus, cloacae, sakazakii) | 21 | 459 | 0 | 480 | 60 |
| Achromobacter xylosoxidans | 1 | 0 | 0 | 1 | 0 |
| Providencia (stuartii, rettgeri) | 1 | 48 | 6 | 55 | 132 |
| Proteus (mirabilis, vulgaris) | 0 | 394 | 53 | 447 | 159 |
| Serratia (marcescens, liquefaciens, odorifera) | 0 | 144 | 4 | 148 | 197 |
| Burkholderia spp. | 0 | 6 | 0 | 6 | 7 |
| Vibrio cholerae | 0 | 24 | 279 | 303 | 180 |
| Morganella (morganii) | 0 | 31 | 19 | 50 | 85 |
| Neisseria (meningitidis, gonorrhoeae) | 0 | 207 | 0 | 207 | 199 |
| Bordetella (bronchiseptica) | 0 | 1 | 2 | 3 | 21 |
| Alcaligenes faecalis | 0 | 3 | 47 | 50 | 0 |
| Stenotrophomonas (maltophilia) | 0 | 5 | 126 | 131 | 6 |
| Leclercia adecarboxylata | 0 | 1 | 0 | 1 | 0 |
| Pantoea (dispersa, agglomerans) | 0 | 2 | 1 | 3 | 21 |
| Mycoplasma genitalium | 0 | 266 | 0 | 266 | 23 |
| Raoultella | 0 | 3 | 0 | 3 | 0 |
| Aeromonas | 0 | 0 | 11 | 11 | 3 |
| Bacillus | 0 | 0 | 97 | 97 | 96 |
| Brevundimonas | 0 | 0 | 3 | 3 | 120 |
| Chromobacterium | 0 | 0 | 6 | 6 | 0 |
| Myroides | 0 | 0 | 2 | 2 | 0 |
| Psychrobacter | 0 | 0 | 3 | 3 | 0 |
| Trabulsiella | 0 | 0 | 1 | 1 | 8 |
| **Total** | **2,551** | **12,568** | **3,832** | **18,951** | **3,028** |

[a]Discrepancies between total isolates in Table 1 and total isolates under Results arise from the fact that not all isolates in every study were Gram-negative isolates, i.e., some studies described resistance in both Gram-negative and Gram-positive bacteria. Non-Gram-negative isolates are not included in Table 1.

seria gonorrhoeae/meningitidis, Proteus mirabilis, and Enterobacter spp., were mostly concentrated (based on the count of each species per country and sample source) in Algeria, Burkina Faso, Egypt, Ghana, Kenya, Libya, South Africa, Tanzania, and Tunisia in humans (Tables S1 to S3). South Africa, Tanzania, and Nigeria reported the highest concentrations of environmental species. Notably, Escherichia coli and S. enterica, and to a lesser extent Campylobacter coli/jejuni, Klebsiella spp., and Pseudomonas spp., were the most often isolated species from animals in the reporting countries. It is interesting that N. gonorrhoeae/meningitidis were mainly reported from humans in Kenya and Niger while Vibrio cholerae/Vibrio spp. were mostly isolated from the environment in South Africa and, to a lesser extent, from humans in Cameroon (Tables S1 to S3); yet, genomes of Neisseria meningitidis were obtained from 10 countries in Southern, Eastern, Western, and Northern Africa (Tables S4 to S6).

E. coli was isolated at very high frequencies in almost all reporting countries (studies and genomes) except Kenya, Ethiopia, Botswana, Zambia, and Senegal (in humans); Ghana, Burkina Faso, and Botswana (in animals); and Nigeria, Egypt, and Cameroon (in the environment). Klebsiella pneumoniae was less common in humans in Egypt, Ethiopia, DRC, Cameroon, Botswana, Benin, Zimbabwe, Zambia, Niger, and Malawi; it was hardly ever reported from animals in Egypt and Cameroon and found only in the environment in Nigeria. S. enterica was mainly distributed in Algeria, Ethiopia, Ghana, Kenya, and Zambia (in humans) and Zambia, Tunisia, South Africa, Kenya, Ethiopia, and Algeria (in animals); it was reported from Egypt only from the environment. In humans, P. aeruginosa was mostly found in Egypt, Burkina Faso, Tanzania, South Africa, and Nigeria while Egypt alone reported it in animals and Nigeria alone reported Pseudomonas spp. in the environment. Interestingly, A. baumannii was mainly concentrated in

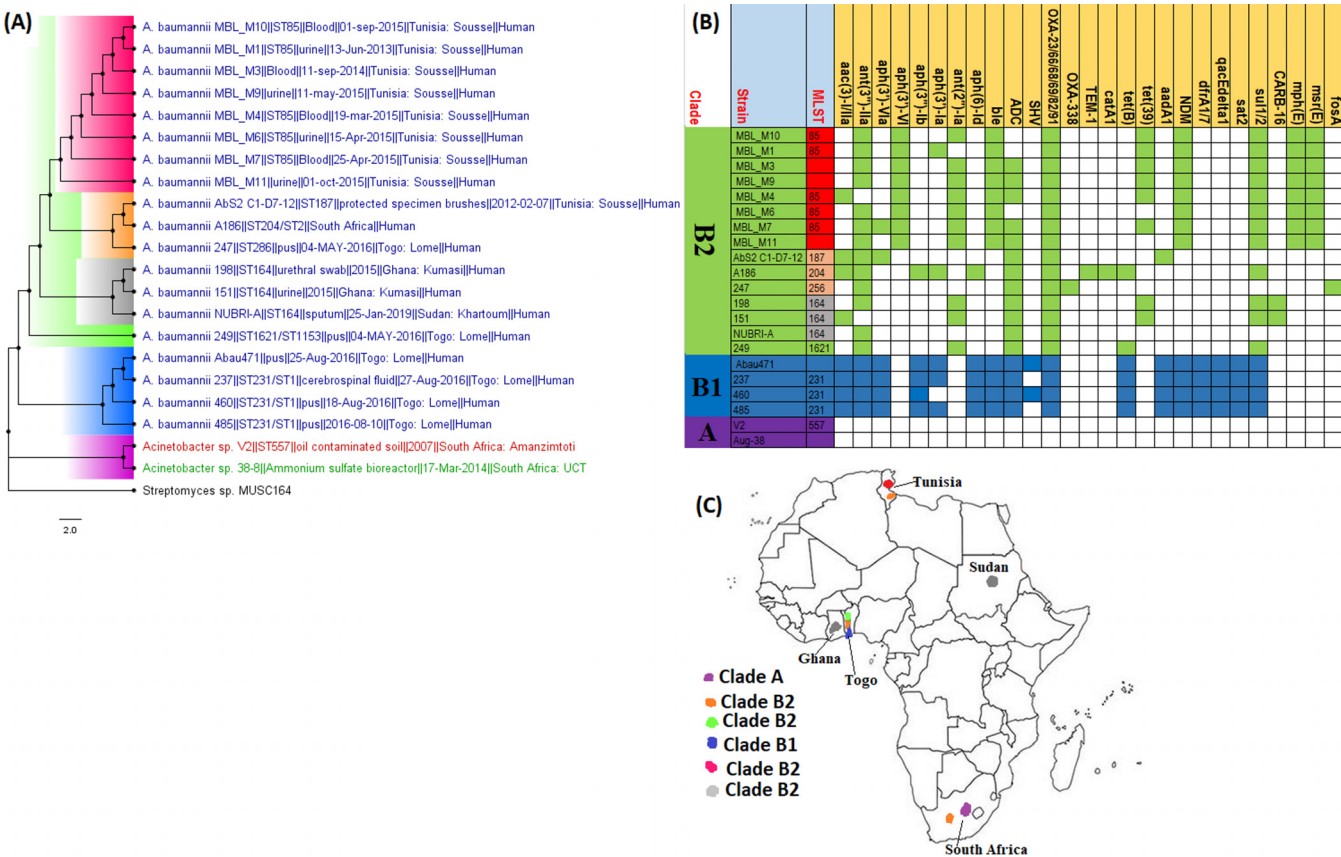

**FIG 2** Geographic distribution of *Acinetobacter baumannii* clades and associated resistomes in Africa. The included *A. baumannii* genomes were mainly from Tunisia, Togo, Ghana, Sudan, and South Africa, with clade-specific ARGs; most of these strains were from humans. Cluster A, which was not *A. baumannii*, had no ARGs, while clusters/clades B1 and B2 had OXA-23-/66-like carbapenemases, *ble*, *ant(2")-Id*, *aph(3')-Ib*, and *ant(3")-IIa*. Isolates from humans, animals, the environment, and plants are colored blue, red, mauve/pink, and green, respectively, on the phylogeny tree.

Ethiopia and Egypt (humans) and in Algeria (animals); *Acinetobacter calcoaceticus/Acinetobacter* species was found from the environment in South Africa and Nigeria (Table S3).

Only *Campylobacter coli/jejuni* had more animal sources than human and environmental sources, and *S. enterica* was the second most frequently isolated species from animal samples after *E. coli*. Notably, *Vibrio cholerae*, *Stenotrophomonas maltophilia*, *Bacillus* spp., *Alcaligenes faecalis*, *Aeromonas* spp., *Chromobacterium* spp., *Brevundimonas* spp., *Psychrobacter* spp., *Myroides* spp., and *Trabulsiella* spp. were either mainly or only found from environmental sources (Table 1).

*Neisseria meningitidis/gonorrhoeae* and *Mycoplasma genitalium*, two sexually transmitted infectious pathogenic species, were mainly found in clinical samples (Table 1; see Fig. 12 and 19 below). However, other *Mycoplasma* spp. were found in chicken (*M. gallinarum/gallinaceum/pullorum*), goat (*Mycoplasma bovis*), cattle (*M. mycoides*), ostrich (*M. nasistruthionis*), sewage (*M. arginini*), and humans (*M. pneumoniae*), and these had no known resistance genes (see Fig. 19 below). Species such as *Burkholderia cepacia/Burkholderia* spp., *Bordetella* spp., *Morganella morganii*, *A. faecalis*, *S. maltophilia*, *Leclercia adecarboxylata*, *Pantoea* spp., and *Raoultella* spp. were rare in clinical samples (Table 1 and Fig. S1 and S3; see also Fig. 21 to 26 below).

**Geographical and host distribution of clones, ARGs, and MGEs.** The clonality of *A. baumannii*, *C. coli/jejuni*, *E. coli*, *K. pneumoniae*, *P. aeruginosa*, *V. cholerae*, *Serratia marcescens*, *N. gonorrhoeae/meningitidis*, and *Salmonella enterica* serovar Enteritidis/Typhi/Typhimurium strains was reported in the various countries out of the 30 species (see Fig. 9 to 11 below). However, only *E. coli* clones, *viz.*, ST38, ST69, ST131, ST410, etc.,

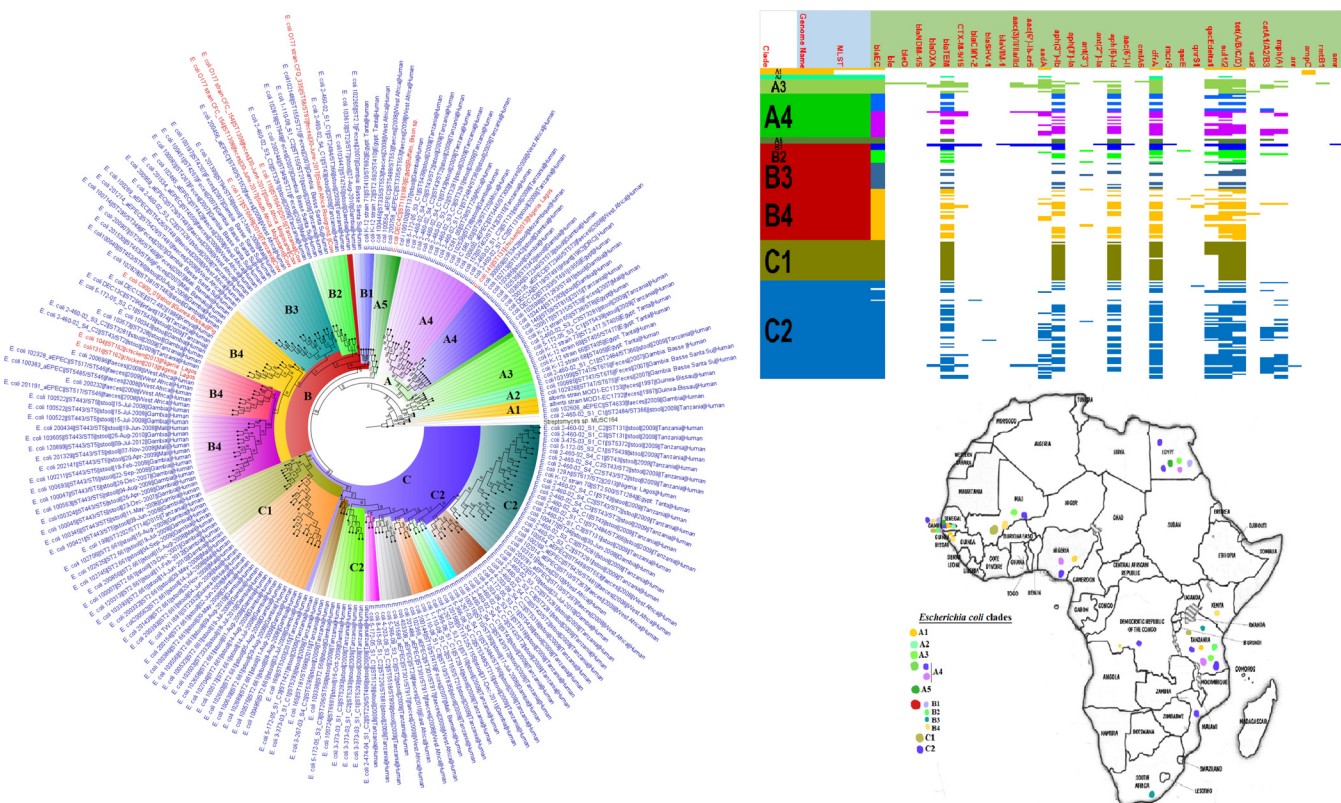

**FIG 3** Geographic distribution of *Escherichia coli* clades (from first 200-genome set) and associated resistomes in Africa. The *E. coli* clades were mostly from humans, mainly distributed in West and East Africa, Egypt, and South Africa (Fig. 3 to 5). Relatively few were from the environment and animals. The ARGs [*tet*(A/B/C/D), *bla*TEM-1, *sul*, *aph(3″)-Ib*, *aph(6)-Id*, and *dfrA*] were mostly conserved across the various clades, which were not region specific but mixed up. Strains from humans shared very close phyletic relationships with strains from animals and plants. Intercountry as well as human-animal dissemination of isolates of the same clade was observed. Isolates from humans, animals, the environment, and plants are colored blue, red, mauve/pink, and green, respectively, on the phylogeny tree.

and groups A/B/C/D were found in humans (in Algeria, B. Faso, CAR, Egypt, Libya, Nigeria, Sao Tome and Principe, Tanzania, Tunisia, and Zimbabwe), animals (Algeria, Egypt, Ghana, Tunisia, and Uganda) and the environment (Algeria and South Africa). Specifically, *E. coli* ST38 was found in humans (Algeria) and animals (Ghana) and groups A/B/D were found in humans (Egypt), animals (Algeria, Egypt, Tunisia, Uganda, and Zimbabwe), and the environment (Algeria and South Africa). Intercountry detection of *E. coli* ST131 in humans was also observed in Algeria, B. Faso, CAR, DRC, Tanzania, Tunisia, and Zimbabwe. *K. pneumoniae* ST101 was also found in Algeria, South Africa, and Tunisia. As well, multiclonal *C. jejuni* strains (i.e., ST19, ST440, ST638, ST9024, etc.) were found in humans and animals from Botswana (Tables S1 to S3).

The clones of the various species from the genomic data did not always agree with those obtained from the included articles in terms of geographical distribution and incidence. For instance, the *E. coli* genomes were highly multiclonal, consisting of 202 clones; the most common of these were ST661, ST10, ST443, ST131, and ST29. *K. pneumoniae* (85 clones) and *S. enterica* (66 clones) genomes were also very multiclonal, with *K. pneumoniae* ST101, ST152, ST15, ST14, ST17, and ST147 and *S. enterica* ST2, ST1, ST198, ST11, ST313, ST321, and ST2235 being very common. Notably, *N. meningitidis* (genomes) ST11, ST2859, ST1, etc., were also common in humans from Ghana (*n* = 63 isolates), B. Faso (*n* = 57 isolates), Niger (*n* = 28 isolates), etc., as seen in the articles. In contrast, *P. aeruginosa* (ST234 and ST235), *A. baumannii* (ST1, ST85, and ST164), *C. jejuni* (ST362), *V. cholerae* (ST69 and ST515), *Bordetella pertussis* (ST1 and ST2 in Kenya), *Mycoplasma pneumoniae* (in Egypt and Kenya), and *Bacillus cereus* and *Bacillus subtilis* (genomes) had relatively few dominant clones (Table S6).

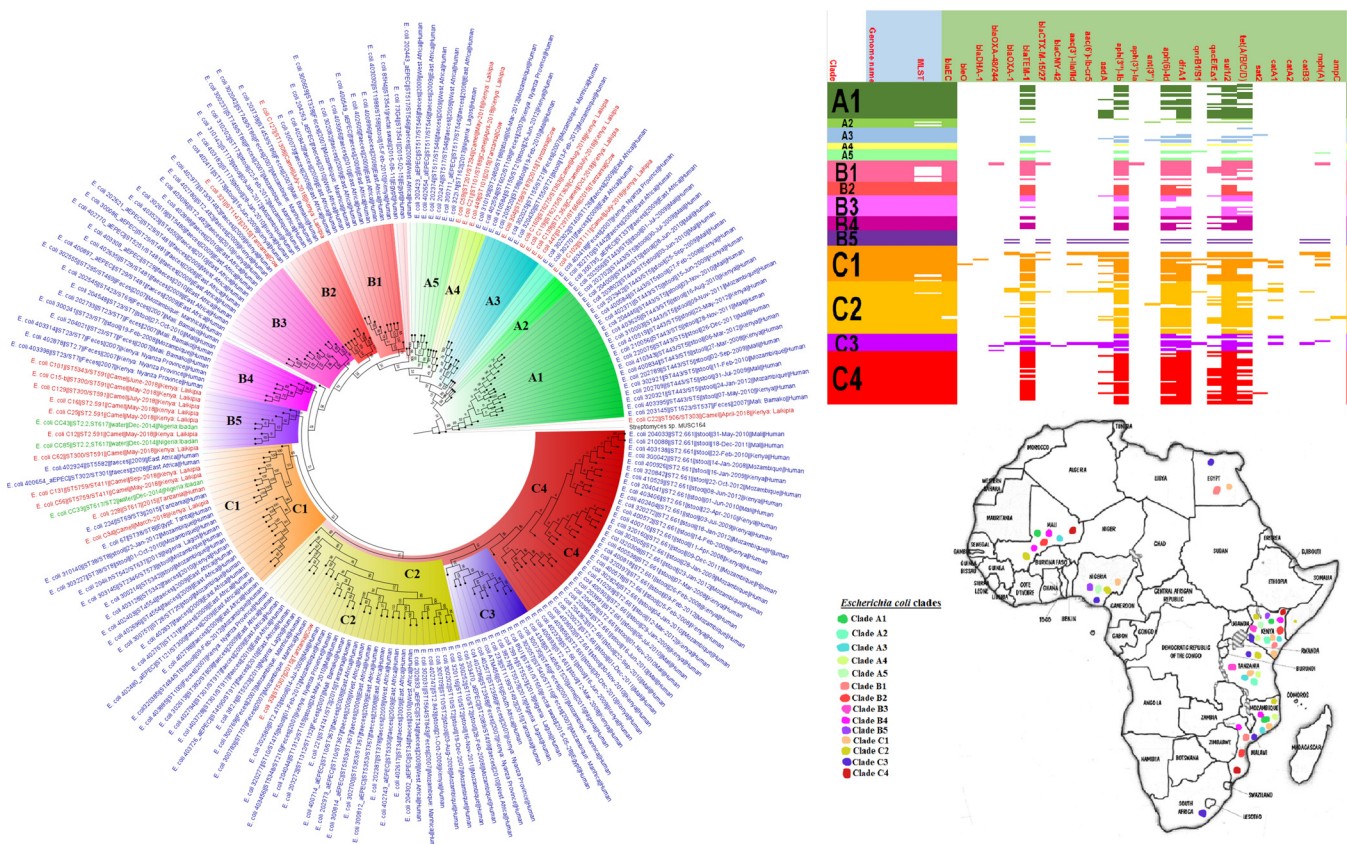

**FIG 4** Geographic distribution of *Escherichia coli* clades (from second 200-genome set) and associated resistomes in Africa. The *E. coli* clades were mostly from humans, mainly distributed in West and East Africa, Egypt, and South Africa (Fig. 3 to 5). The ARGs [*tet*(A/B/C/D), *bla*$_{TEM-1}$, *sul*, *aph(3″)-Ib*, *aph(6)-Id*, and *dfrA*] were mostly conserved across the various clades, which were not region specific but mixed up. Strains from humans shared very close phyletic relationships with strains from animals and plants. Intercountry as well as human-animal dissemination of isolates of the same clade was observed. Isolates from humans, animals, the environment, and plants are colored blue, red, mauve/pink, and green, respectively, on the phylogeny tree.

ARGs mediating resistance to almost all known Gram-negative bacterial antibiotics were found in the included articles, with more of these ARGs being isolated from human strains rather than animal and environmental species in a descending order. Notably, ARGs conferring resistance to β-lactams, specifically ESBLs such as CTX-M, TEM, SHV, OXA, and GES and AmpCs such as CMY, FOX, DHA, MOX, ACC, EBC, and LEN, were commonly identified in human, animal, and environmental isolates from most countries, with *bla*$_{CTX-M}$ and *bla*$_{TEM}$ being the most frequently identified ARGs. Moreover, OXA and GES ESBLs and all the AmpCs as well as carbapenemases (i.e., OXA-48/181/204, OXA-23/51/53, NDM, IMP, SPM, VIM, KPC, and GES-5) were not reported from animal or environmental isolates; only OXA-61 (from *C. jejuni*) was found in animal isolates in Botswana. Carbapenemase genes were relatively less often detected in human strains and reported from a few countries: the metallo-β-lactamases such as NDM, IMP, SPM, and VIM were mainly found in Egypt, South Africa, Tanzania, Tunisia, and Uganda; KPC and GES-5 were common in South Africa and Uganda; and the OXA types were found in Algeria, Egypt, Nigeria, Sao Tome and Principe, South Africa, Tunisia, and Uganda (Tables S1 to S3).

Second to the β-lactams, there was frequent detection of diverse fluoroquinolone resistance mechanisms in human and animal isolates from almost all the countries: *aac(6′)-Ib-cr*, *aac(3′)-IIa*, *aac(3′)-Ih*, *qnrA/B/D/S*, *oqxAB*, and chromosomal mutations in *gyrAB*, *parCE*, and *qepA* in a descending order. None of these mechanisms were found in environmental strains from the included studies. Moreover, aminoglycoside resistance mechanisms, including *aac(6′)-Ib-cr*, which also confers resistance to fluoroquinolones, were equally highly distributed in human and animal isolates, with relatively

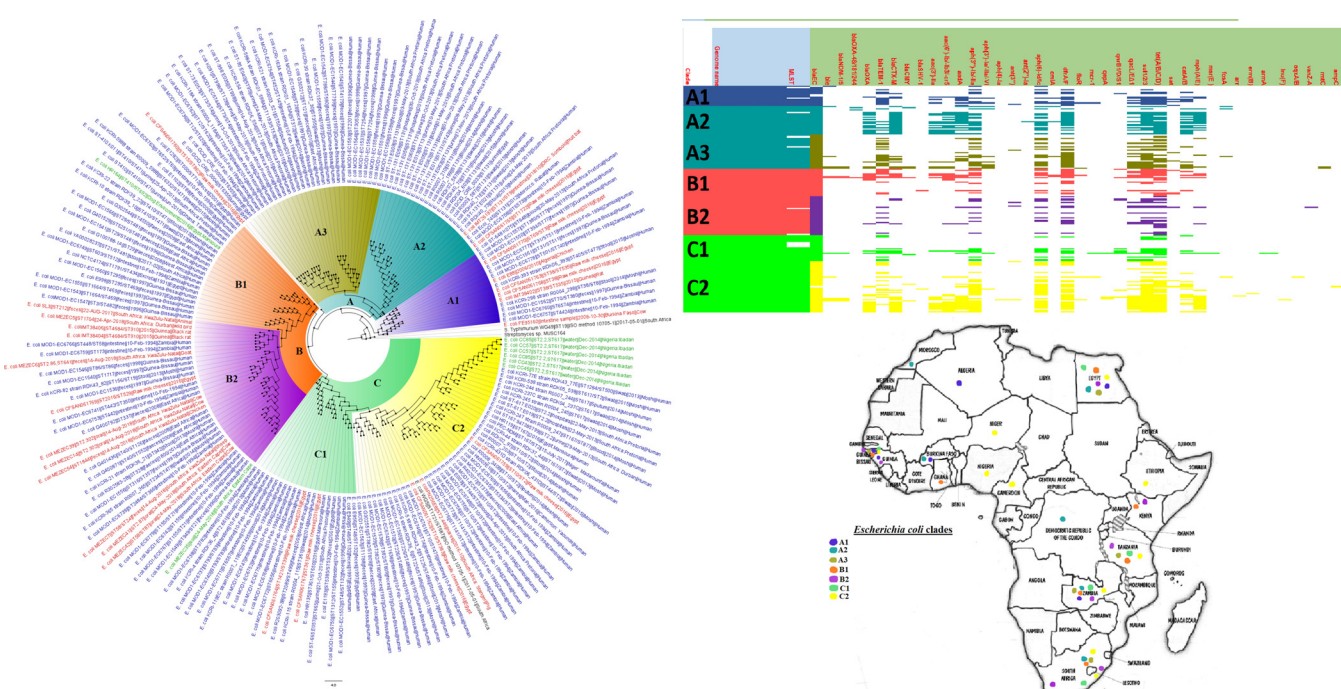

**FIG 5** Geographic distribution of *Escherichia coli* clades and associated resistomes in Africa (from third 200-genome set). The *E. coli* clades were mostly from humans, mainly distributed in West and East Africa, Egypt, and South Africa. Few genomes were from the environment and animals. The ARGs [*tet*(A/B/C/D), *bla*$_{TEM-1}$, *sul*, *aph(3″)-Ib*, *aph(6)-Id*, and *dfrA*] were mostly conserved across the various clades, which were not region specific but mixed up. Strains from humans shared very close phyletic relationships with strains from animals and plants. Intercountry as well as human-animal dissemination of isolates of the same clade was observed. Isolates from humans, animals, the environment, and plants are colored blue, red, mauve/pink, and green, respectively, on the phylogeny tree.

limited occurrence in environmental strains. Among these aminoglycoside mechanisms were *aadA*, *strAB*, *aph(3′)*, *aph(6′)*, *ant(2′)*, *ant(3′)*, and the 16S rRNA methyltransferases such as *rmtC/F* and *armA*.

Other common resistance mechanisms that were highly distributed in almost all strains from almost all the included countries were *sul1/2/3* and *dfrA* (mediating resistance to sulfamethoxazole-trimethoprim); these were mostly found in animal and human strains and relatively less often isolated from environmental strains. Chloramphenicol resistance genes, *viz.*, *cmlA/B* and *catA/B*, were also found in animal and human isolates in substantial numbers while ARGs for florfenicol (*floR*) and fosfomycin (*fosA*) were very rare, being found in only human strains. Of note, tetracycline ARGs, *tet*(O/A/B/C/D/G/K/M39), were almost fairly distributed in strains from humans (H), animals (A), and the environment (E) in relatively few countries such as Algeria (E), Botswana (A and H), Cameroon (E), Ethiopia (A), Malawi (H), Nigeria (E), South Africa (A, H, and E), Tanzania (A and E), Tunisia (A and H), Uganda (A), and Zambia (A).

Interestingly, colistin resistance mechanisms such as *mcr-1* and chromosomal mutations in *pmrAB* were very rare. Particularly, *pmrAB* mutations were recorded only in human strains from Tunisia while *mcr-1* genes were reported only in South Africa (A, H, and E), Sao Tome and Principe (H), and Tunisia (A). Other rare ARGs, found mainly in human isolates, included *blaZ*, *pse-1*, and *penA* (conferring penicillin resistance), *ermABC* and *mph(A)* (encoding erythromycin/macrolide resistance), *cmeAB* (multidrug efflux system in *Campylobacter jejuni*), *porAB* (porin in *Neisseria* spp.), *macA/B* (encoding part of the tripartite efflux system MacAB-TolC for transporting macrolides from the cytosol), *qacEΔ1* (encoding resistance to quaternary ammonium compounds through efflux), and *mexAB* (encoding multidrug resistance [MDR] efflux pumps in *Pseudomonas* spp.).

These ARGs were found associated with MGEs such as plasmids, integrons, insertion sequences (ISs), and transposons, mainly in human isolates; MGEs in animal and environmental isolates were rarely described (Tables S1 to S3). The most common MGEs were IncF-type, IncX$_3$, ColRNAI, IncR, IncY, IncL/M, A/C, IncH, and IncQ plasmids, *IS*EcP1

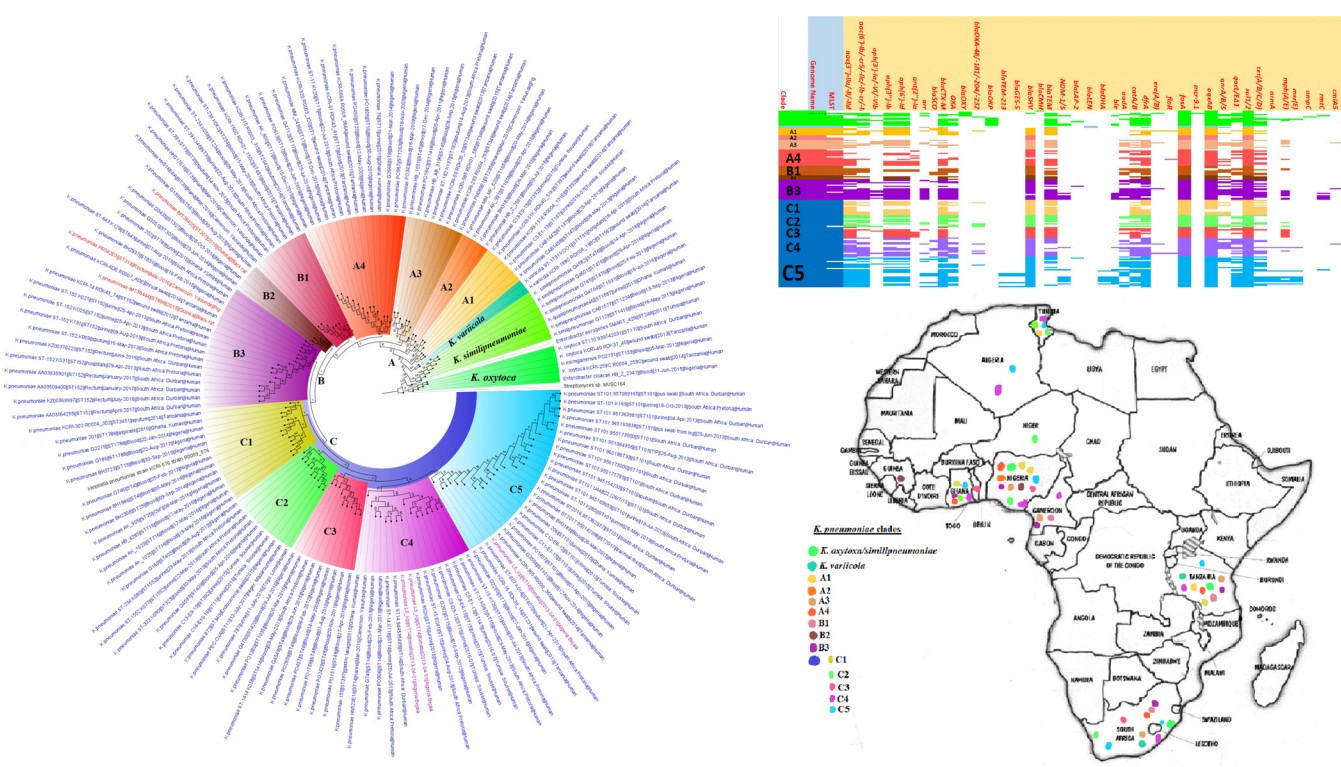

**FIG 6** Geographic distribution of *Klebsiella pneumoniae* clades (from first 200-genome set) and associated resistomes in Africa. *Klebsiella pneumoniae* was mainly from humans, with a few strains being isolated from plants and animals. The various clades were mixed up in South Africa, Tanzania, Uganda, and West and North Africa. Strains from humans shared very close phyletic relationships with strains from animals and plants (Fig. 6 and 7). The clades harbored many conserved ARGs (*n* = 14): *aac(3')-Ia/IIa*, *aac(6')-IIa/Ib-cr*, *aph(3")-Ib*, *aph(6')-Id*, *bla*$_{CTX-M}$, *bla*$_{OXA}$, *bla*$_{SHV}$, *bla*$_{TEM}$, *aadA*, *catA/B*, *dfrA*, *fosA*, *oqxAB*, and *sul1/2*. Other ARGs that were substantially found in *K. pneumoniae* included *arr*, *mph(A/E)*, *qnrA/B/S*, *qacE*Δ1, and *bla*$_{NDM-1/5}$. A few genes were restricted to certain clades. Intercountry as well as human-animal dissemination of isolates of the same clade was observed. Isolates from humans, animals, the environment, and plants are colored blue, red, mauve/pink and green, respectively, on the phylogeny tree.

and class 1 integrons (IntI1). ColE, IncU, IncLVPK, IncP, IncI, IncN, ColpVc, and ColKp3 plasmids, *Tn2006*, *IS*Kpn19, *Tn3*, *IS26*, *Tn21*, and *IS*AbaI transposons and ISs were less frequently identified MGEs. Specifically, IncF-type plasmids were commonly associated with ESBLs and carbapenemase genes while IntI1 was common with both β-lactamases and non-β-lactamase genes such as *sul1/2/3*, *dfrA*, *catA/B*, *floR*, *qepA*, and *qnrA/B/D/S*. *IS*EcP1 was common around *bla*$_{CTX-M-15}$, *Tn206* bracketed *bla*$_{OXA-23/51}$ and *aac(3')-I*, and *Tn402*-like transposon harbored *bla*$_{VIM-5}$, *aph(3')-Ib*, *aph(6')-Id*, *tet*(C/G), and *floR* (in environmental strains in Nigeria) (Tables S1 to S3).

**Resistance levels.** Rates of resistance to the various antibiotics were highest among human strains, particularly in Egypt, Ethiopia, Mali, Senegal, Tunisia, and Uganda among *Enterobacteriaceae* such as *E. coli*, *K. pneumoniae*, *Salmonella enterica*, *Providencia rettgeri/stuartii*, *Neisseria meningitidis/gonorrhoeae*, *P. aeruginosa*, and *A. baumannii*. These human strains had higher rates of resistance to almost all the antibiotic classes including the aminoglycosides, β-lactams, fluoroquinolones, tetracyclines, sulfamethoxazole-trimethoprim (SXT), and phenicols. Comparatively, strains from animals and the environment had lower resistance rates. Specifically, *A. baumannii*, *E. coli*, *Salmonella enterica*, and *Providencia* spp. were resistant to ampicillin, amikacin, chloramphenicol, kanamycin, tetracycline, streptomycin, sulfonamide, and SXT in most of the included countries. Notably, the rates of resistance of environmental *E. coli*, *S. enterica*, *K. pneumoniae/oxytoca*, and *Citrobacter freundii/koseri* strains to fluoroquinolones, tetracycline, sulfonamide, SXT, and ceftriaxone were substantially high in Algeria, Benin, and Egypt (Tables S1 to S3).

**Phylogenomic and AR gene analyses: evolutionary epidemiology of resistance.** Phylogenetically, strains belonging to the same clades were found in different coun-

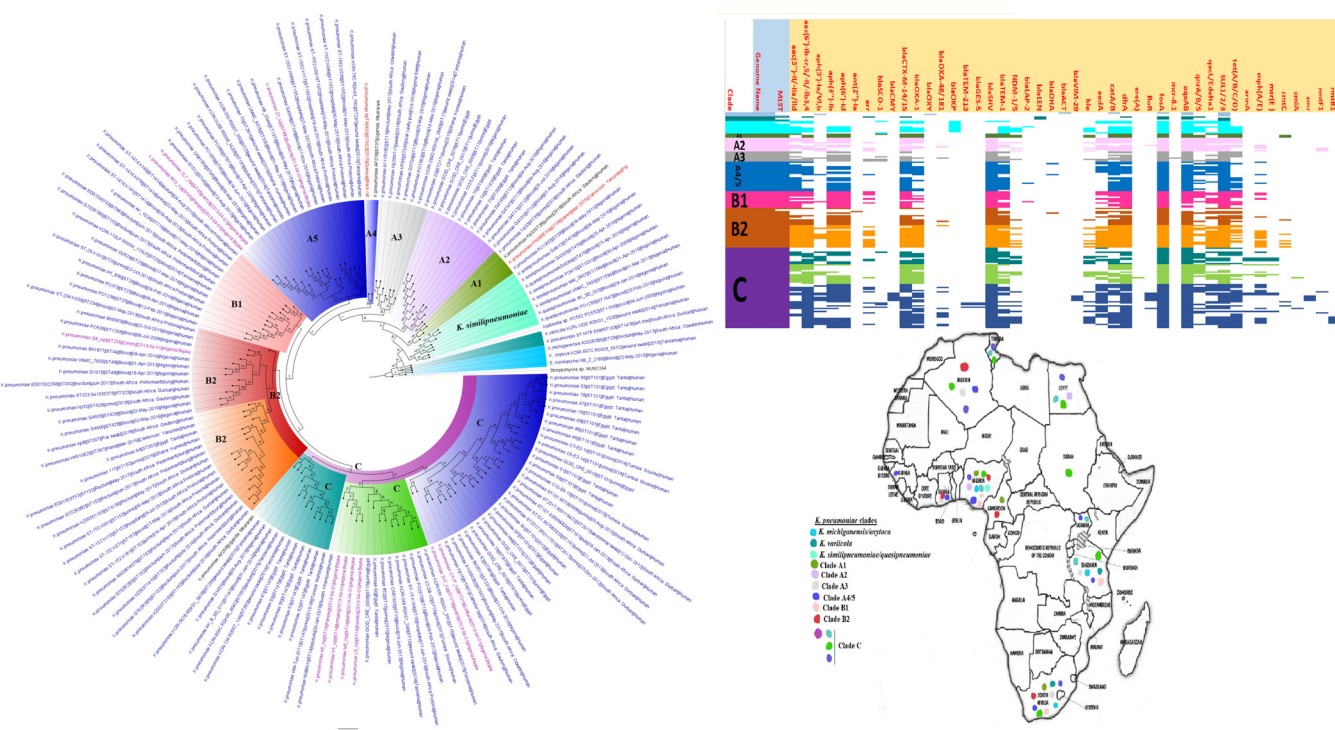

**FIG 7** Geographic distribution of *Klebsiella pneumoniae* clades (from second 200-genome set) and associated resistomes in Africa. *Klebsiella pneumoniae* was mainly from humans, with a few strains being isolated from plants and animals. There was a mixture of the clades in South Africa, Tanzania, Uganda, and West and North Africa. Strains from humans shared very close phyletic relationships with strains from animals and plants (Fig. 6 and 7). The clades harbored many conserved ARGs (*n* = 14): *aac(3')-Ia/IIa*, *aac(6')-IIa/Ib-cr*, *aph(3")-Ib*, *aph(6')-Id*, *bla*CTX-M, *bla*OXA, *bla*SHV, *bla*TEM, *aadA*, *catA/B*, *dfrA*, *fosA*, *oqxAB*, and *sul1/2*. Other ARGs that were substantially found in *K. pneumoniae* included *arr*, *mph(A/E)*, *qnrA/B/S*, *qacEΔ1*, and *bla*NDM-1/5. A few genes were restricted to certain clades. Intercountry as well as human-animal dissemination of isolates of the same clade was observed. Isolates from humans, animals, the environment, and plants are colored blue, red, mauve/pink, and green, respectively, on the phylogeny tree.

tries, and in a limited measure, in humans, animals, the environment, and plants. Among the species, certain countries contained only a single clade of a species while some countries contained several clades of the same species: *E. coli* (Algeria), *K. pneumoniae* (Mali), *N. meningitidis* (South Africa and DRC), *Campylobacter* spp. (South Africa), etc. (Fig. 2 to 19). Within specific clades were found, in a few cases, isolates from different sample sources: clades A, B, and C (*E. coli*), clade B (*P. aeruginosa*), etc. (Fig. 3 to 7). Notably, strains belonging to different multilocus sequence types (MLSTs) were found within the same clades. Generally, the genomes of included species were from Southern, Eastern, Western, and Northern Africa, with little or none from Central Africa; countries reporting the most genomes included Ghana, Mali, Nigeria, Cameroon, Tunisia, Algeria, Egypt, Kenya, Tanzania, Mozambique, and South Africa (Fig. 3 to 26 and Fig. S1 to S3).

*Enterobacter* spp. and *K. pneumoniae* strains (put together) contained the largest and richest repertoire (collection) of AR genes compared to the other species (Fig. 1 and 3 to 7; Table S7). In a descending order, *S. marcescens*, *S. enterica*, *E. coli*, *A. baumannii*, *P. aeruginosa*, *V. cholerae*, *Citrobacter freundii*, *Providencia rettgeri*, *Proteus mirabilis*, and *M. morganii* strains harbored a rich and diverse repertoire of ARGs. No known ARGs were found in the other species, although the literature reported resistance mechanisms in *N. meningitidis* (*gyrA*, *penA*, and *rpoB*), *N. gonorrhoeae* (*gyrA*, *penA*, *ponA*, *mtrR*, *porB*, and *bla*TEM), *C. jejuni/coli* [*bla*OXA-61, *tet*(O), *gyrA*(T86I), *cmeABC*, and *aph(3)-I*], *M. genitalium* (*gyrA* and *parC*), and *S. maltophilia* (*sul3*) (Tables S1 to S3). Some ARGs were conserved within specific clades and species: NDM-1 (*S. marcescens*), OXA-23/66-like (*A. baumannii*), TEM-1 (*E. coli* and *K. pneumoniae*), CTX-M (*Enterobacter* spp. and *K. pneumoniae*), SUL1/2/3 (*Enterobacter* spp., *E. coli*, *K. pneumoniae*, and *S. enterica*), *dfrA* (*Enterobacter* spp., *E. coli*, *K. pneumoniae*, and *S. enterica*), QnrB/S (*Enterobacter* spp., and *K. pneu-*

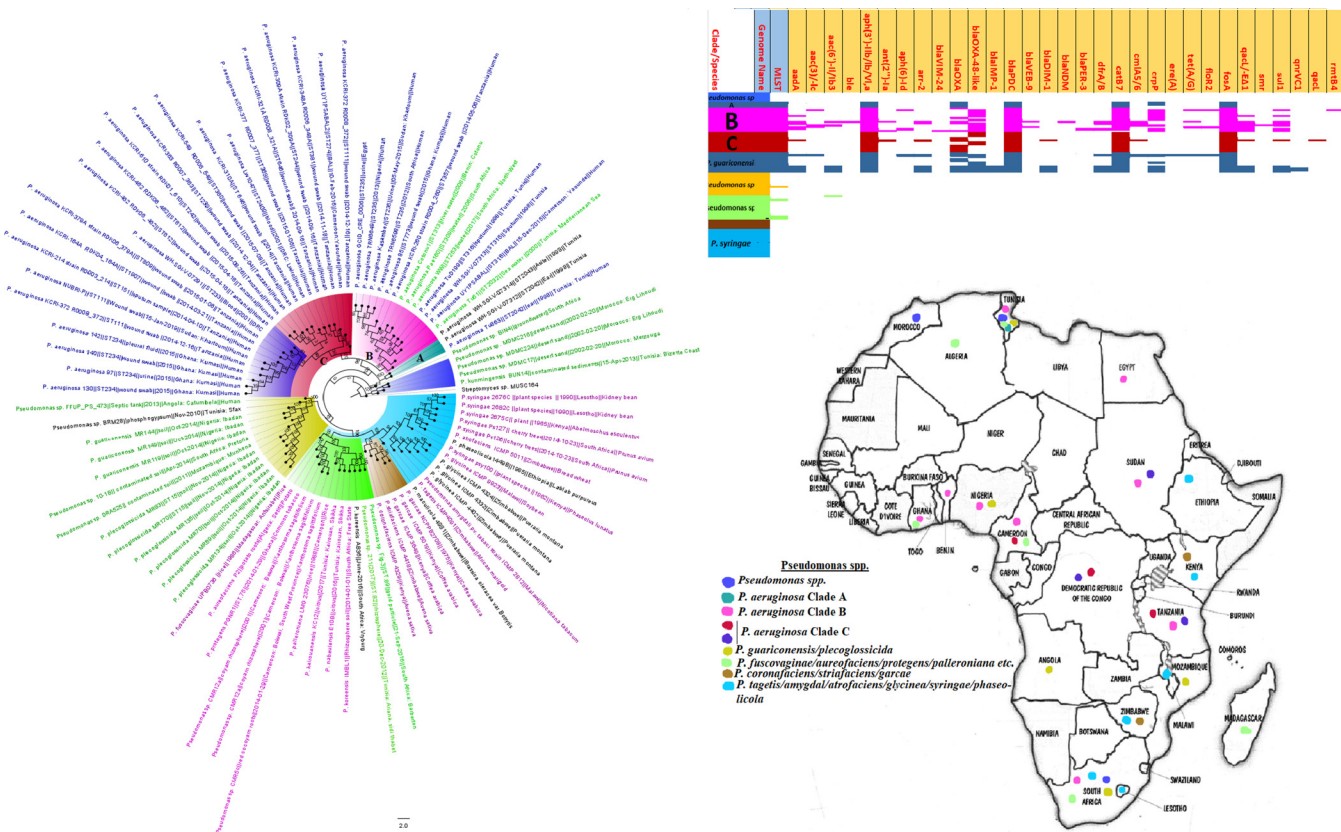

**FIG 8** Geographic distribution of *Pseudomonas* clades and associated resistomes in Africa. *Pseudomonas* spp. were widely distributed in Africa, particularly Southern and Eastern Africa, DRC, and West (Cameroon, Nigeria, Benin, and Ghana) and North (Morocco, Algeria, Tunisia, and Egypt) Africa. The other *Pseudomonas* spp. were mainly found in the environment and plants with no ARGs while *P. aeruginosa* was found in humans and the environment with *aph(3')-IIb*, *bla*$_{OXA}$, *bla*$_{PDC}$, *catB7*, and *fosA* being largely conserved in this pathogen. Clade B had additionally conserved genes such as *sul1/2* and *qacL/ΔE1*. Intercountry as well as human-environment dissemination of isolates of the same clade was observed. Isolates from humans, animals, the environment, and plants are colored blue, red, mauve/pink, and green, respectively, on the phylogeny tree.

*moniae*), TET (*Enterobacter* spp., *E. coli*, *N. meningitidis*, *K. pneumoniae*, and *S. enterica*), etc. (Fig. 3 to 26 and Fig. S1 to S3).

**Diagnostics.** Phenotypic and molecular methods were employed by the included studies to determine the species identity, antibiotic sensitivity (AST), genotype/clone, and resistance mechanisms of the isolates. Broth microdilution (BMD) and disc diffusion methods were common phenotypic tests used for determining the AST of the isolates. Vitek, Etest, and agar dilution methods were less often used. PCR or PCR-based typing methods such as multilocus sequence typing (MLST), repetitive element PCR (REP), and enterobacterial repetitive intergenic consensus (ERIC)-PCR were more commonly used to determine the clonality of enterobacterial isolates than non-PCR-based techniques such as pulse-field gel electrophoresis (PFGE), which is more laborious. Finally, PCR was the most common tool used for determining the ARGs of the isolates, with the use of whole-genome sequencing (WGS) being limited (Table S7).

## DISCUSSION

The AR gene dynamics or epidemiology, phylogenomics, and geographic location of Gram-negative bacterial species and associated clones, MGEs, and ARGs in Africa are here limned for the first time. We show that *E. coli*, *K. pneumoniae*, *S. enterica*, *A. baumannii*, *P. aeruginosa*, *N. meningitidis/gonorrhoeae*, *V. cholerae*, *S. marcescens*, *Enterobacter* spp., *C. jejuni*, *Mycoplasma* spp., *Providencia* spp., *Proteus mirabilis*, and *Citrobacter* spp. are major pathogenic species with rich and diverse AR genes and mobilomes, circulating in humans, animals, and the environment in Southern, Eastern, Western, and Northern Africa. The poorer sanitary conditions and food insecurity as well

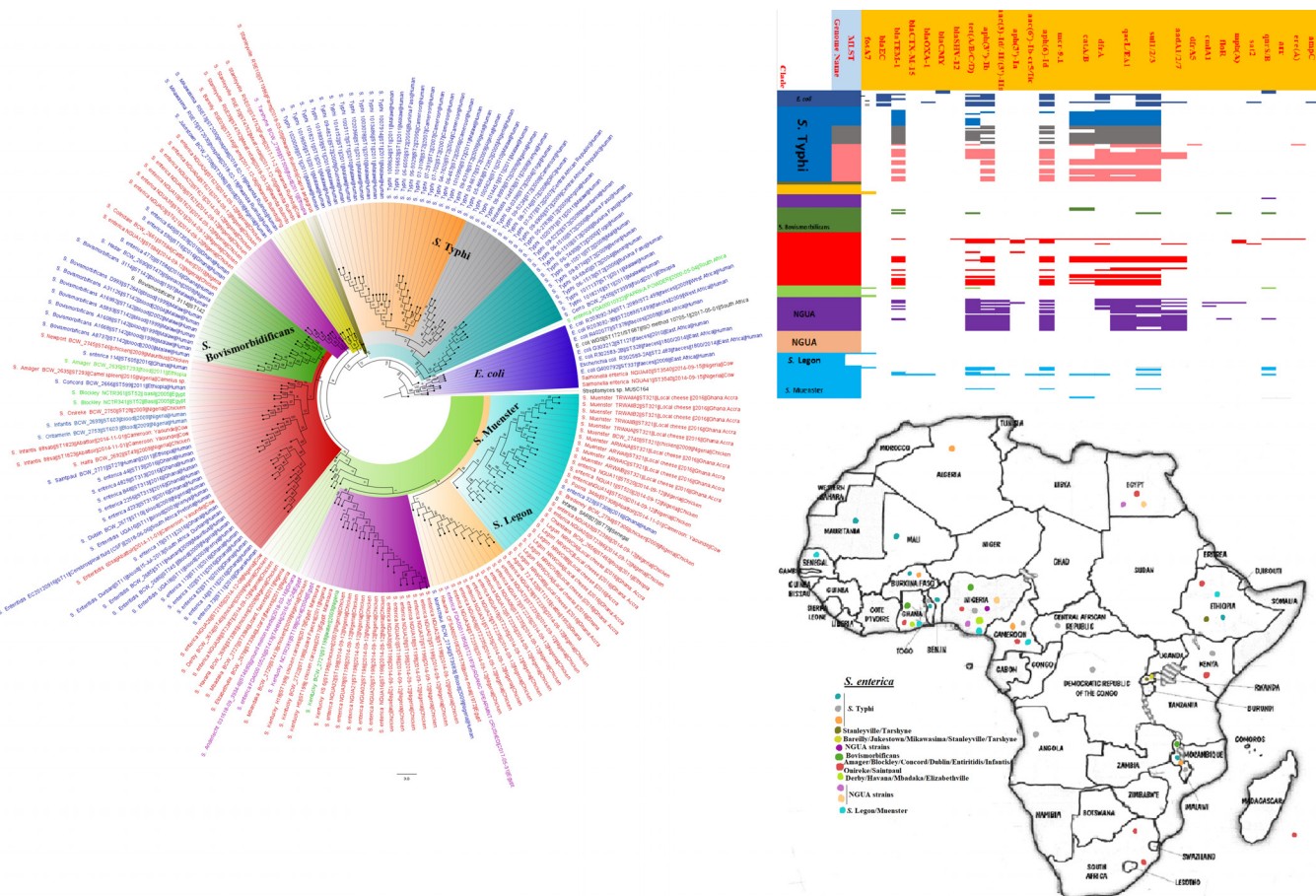

**FIG 9** Geographic distribution of *Salmonella enterica* clades (from first 200-genome set) and associated resistomes in Africa. *Salmonella enterica* serovars were mainly host specific, with *S.* Typhi, *S.* Typhimurium, *S.* Enteritidis, and *S.* Bovismorbificans being isolated from humans while strains of serovars such as *S.* Muenster, *S.* Legon, NGUA, *S.* Salamae, and *S.* Wilhelmsburg were animal associated. In general, *S. enterica* clades were of diverse geographic distribution but clustered in Southern Africa, Madagascar, DRC, Sudan, Comoros, Cameroon, and East, West, and North Africa (Fig. 9 to 11). The ARGs in *S. enterica* were serovar and clade specific, with *S.* Typhi, *S.* Typhimurium, and *S. enterica* NGUA strains hosting most ARGs such as TEM, *tet*(A/B/C/D), *aph(6')-ld*, *aph(3")-lb*, *catA/B*, *dfrA*, *qacL/ΔE1*, and *sul1/2/3*; notably, *S.* Typhi clades A1 and B1 to B3 (Fig. 9 to 11) mostly harbored these ARGs. Intercountry as well as human-animal-environment dissemination of isolates of the same clade was observed. Isolates from humans, animals, the environment, and plants are colored blue, red, mauve/pink, and green, respectively, on the phylogeny tree.

as weaker health care and diagnostic laboratory capacities in Africa are well known, accounting for the higher infectious disease rates on the continent (2–5, 18, 26). Subsequently, the combination of diverse ARGs in highly pathogenic species with wide geographical distribution on the continent is a cause for concern as it provides fertile breeding grounds for periodic and large outbreaks with untold morbidities and mortalities (27–30).

Although *E. coli* was the most frequently isolated species in human, animal, and environmental samples, it did not harbor the most diverse and richest AR gene repertoire (collection of ARGs in all the *E. coli* isolates) (Fig. 1C; see also Table S7 in the supplemental material). This is interesting as *E. coli* is mostly used as a sentinel organism to study Gram-negative bacterial resistance epidemiology (18, 31). Whereas the higher numbers of *E. coli* strains obtained in Africa could be due to the species' easy cultivability, identification, and exchange of resistance determinants (17, 32), as well as its common use as a sentinel organism (18, 31), its lower AR gene diversity and richness could mean it is not representative of the actual AR gene circulating in any niche at a given point in time. Hence, *Enterobacter cloacae/Enterobacter* spp. and *K. pneumoniae*, which contained richer AR genes, could serve as better representatives and reporters of prevailing ARGs in any niche at a point in time. Particularly, the richer AR genes of these

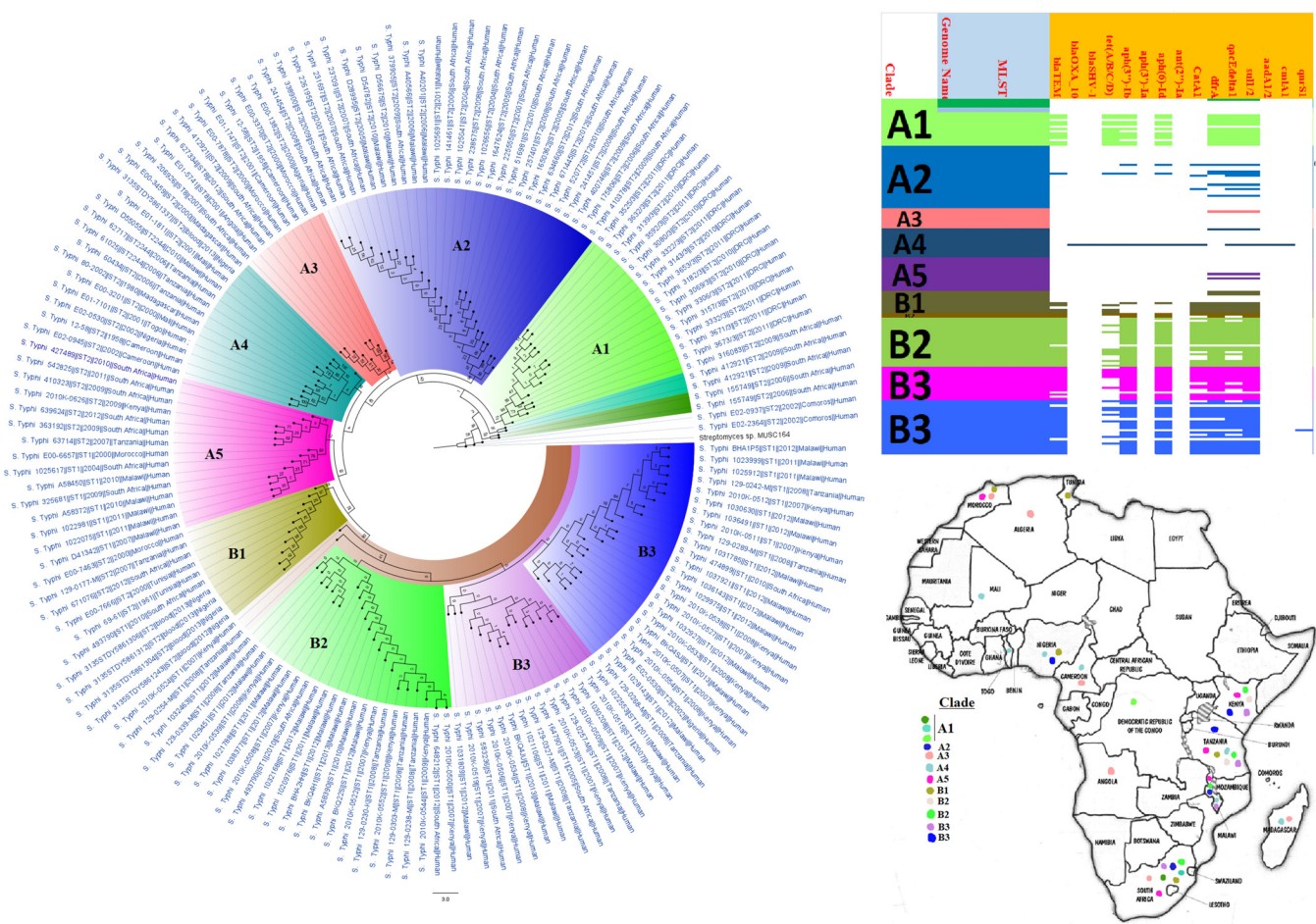

**FIG 10** Geographic distribution of *Salmonella enterica* clades (from second 200-genome set) and associated resistomes in Africa. *Salmonella enterica* serovars were mainly host specific, with *S.* Typhi, *S.* Typhimurium, *S.* Enteritidis, and *S.* Bovismorbificans being isolated from humans. In general, *S. enterica* clades were of diverse geographic distribution but clustered in Southern Africa, Madagascar, DRC, Sudan, Comoros, Cameroon, and East, West, and North Africa (Fig. 9 to 11). The ARGs in *S. enterica* were serovar and clade specific, with *S.* Typhi, *S.* Typhimurium, and *S. enterica* NGUA strains hosting most ARGs such as TEM, *tet*(A/B/C/D), *aph(6′)-Id*, *aph(3″)-Ib*, *catA/B*, *dfrA*, *qacL/ΔE1*, and *sul1/2/3*; notably, *S.* Typhi clades A1 and B1 to B3 (Fig. 9 to 11) mostly harbored these ARGs. Intercountry as well as human-animal-environment dissemination of isolates of the same clade was observed. Isolates from humans, animals, the environment, and plants are colored blue, red, mauve/pink, and green, respectively, on the phylogeny tree.

two species strongly suggest that they can easily exchange ARGs between themselves and other species, as reported already (17, 32–34).

The higher AR gene diversity, as well as the high isolation rates, of *K. pneumoniae* and *Enterobacter* spp. is not surprising. Specifically, *K. pneumoniae* is the most frequently isolated clinical bacterial pathogen in many countries worldwide, found to be involved in many fatal and multidrug-resistant infections (27, 30, 35, 36). International clones such as *K. pneumoniae* ST208 and ST101 are implicated in the clonal dissemination of carbapenemases as well as colistin and multidrug resistance (27, 30, 35–38); although ST208 was absent in Africa, ST101 was common in several countries. As well, *Enterobacter* spp. are increasingly being isolated from many clinical infections in which they are found to be a major host of *mcr* colistin resistance genes and other clinically important MDR determinants (36, 39–41). Notwithstanding their lower AR gene diversity compared to *Enterobacter* spp. and *K. pneumoniae*, the *E. coli* isolates contained important ARGs, such as *mcr-1*, *bla*$_{NDM-1}$, *bla*$_{OXA-48/181}$, and *bla*$_{CTX-M-15}$ (Tables S1 to S3), which can be transferred to other intestinal pathogens (16, 17, 32). Notably, the *E. coli* isolates also exhibited high rates of resistance to important clinical antibiotics (Tables S1 to S3). Finally, the *Enterobacter* species, *K. pneumoniae*, and *E. coli* strains were generally highly multiclonal and evolutionarily distant, suggesting little clonal dissemination (except *E. coli* ST103 and *K. pneumoniae* ST101) of prevalent clones within these

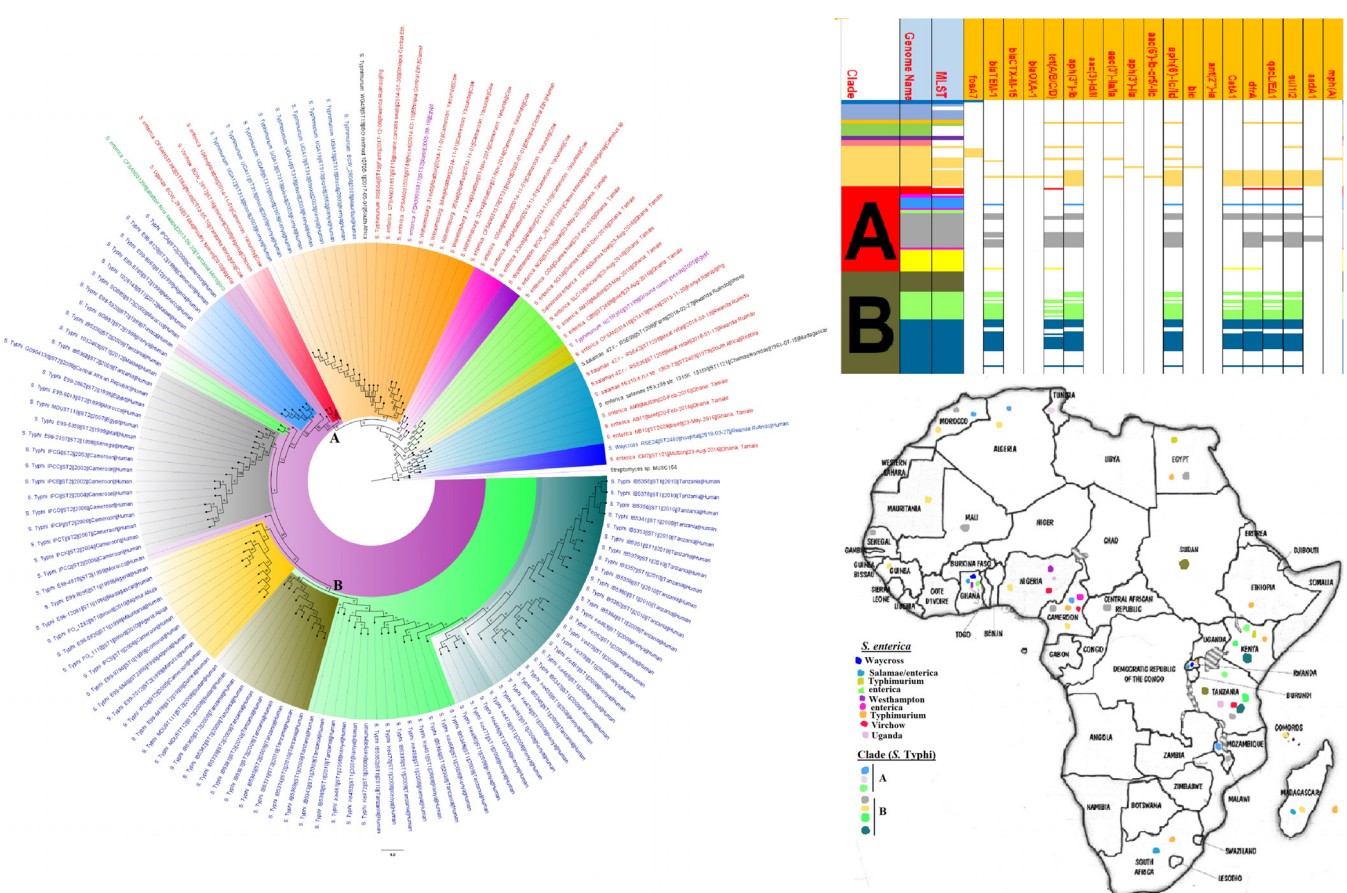

**FIG 11** Geographic distribution of *Salmonella enterica* clades (from third 200-genome set) and associated resistomes in Africa. *Salmonella enterica* serovars were mainly host specific, with strains of serovars such as *S.* Muenster, *S.* Legon, NGUA, *S.* Salamae, and *S.* Wilhelmsburg being animal associated. In general, *S. enterica* clades were of diverse geographic distribution but clustered in Southern Africa, Madagascar, DRC, Sudan, Comoros, Cameroon, and East, West, and North Africa (Fig. 9 to 11). The ARGs in *S. enterica* were serovar and clade specific, with *S.* Typhi, *S.* Typhimurium, and *S. enterica* NGUA strains hosting most ARGs such as TEM, *tet*(A/B/C/D), *aph(6')-Id*, *aph(3")-Ib*, *catA/B*, *dfrA*, *qacL/ΔE1*, and *sul1/2/3*; notably, *S.* Typhi clades A1 and B1 to B3 (Fig. 9 to 11) mostly harbored these ARGs. Intercountry as well as human-animal-environment dissemination of isolates of the same clade was observed. Isolates from humans, animals, the environment, and plants are colored blue, red, mauve/pink, and green, respectively, on the phylogeny tree.

species across the continent, albeit local and limited intercountry outbreaks were observed (Fig. 3 to 7 and 14) (35, 42).

*S. enterica* and *C. coli/jejuni*, which are important zoonotic and foodborne pathogens (26, 43), were found in animal/food, human, and environmental samples, although *S. enterica* was more common and had a higher AR gene diversity than *C. coli/jejuni* (Table 1 and Tables S1 to S3; Fig. 9 to 11 and 16). Notably, *S.* Typhi, *S.* Typhimurium, *S.* Enteritidis, and *S.* Bovismorbificans were mostly isolated from humans, with some *S.* Enteritidis and *S.* Infantis strains being isolated from both humans and cattle (Fig. 9 to 11). Indeed, reports of *S.* Typhimurium and *S.* Enteritidis isolation from pigs and poultry, respectively, as well as their implication in fatal zoonotic infections through contaminated food animal consumption, are well documented (26, 44–47).

*S.* Typhi, a common foodborne pathogen that infects millions of people worldwide annually and results in typhoid fever, diarrhea, and death in severe cases (18, 20, 21, 48), was the third most common species to be isolated and the fourth species to host the largest AR gene repertoire. As shown in Tables S4 to S6 and Fig. 9 to 11, several isolates from Eastern, Southern, and Western Africa shared the same clone (ST1 and ST2) and clade, representing clonal outbreaks affecting many people over a large swath of Africa. This is observed in Fig. 9 to 11 in countries such as Tanzania (2006 and 2009 to 2010), Malawi (2010 to 2012), Nigeria (2013), and Kenya (2008). Moreover, *S.* Typhi and other *S. enterica* serovars recorded very high levels of resistance to first-line antibiotics in

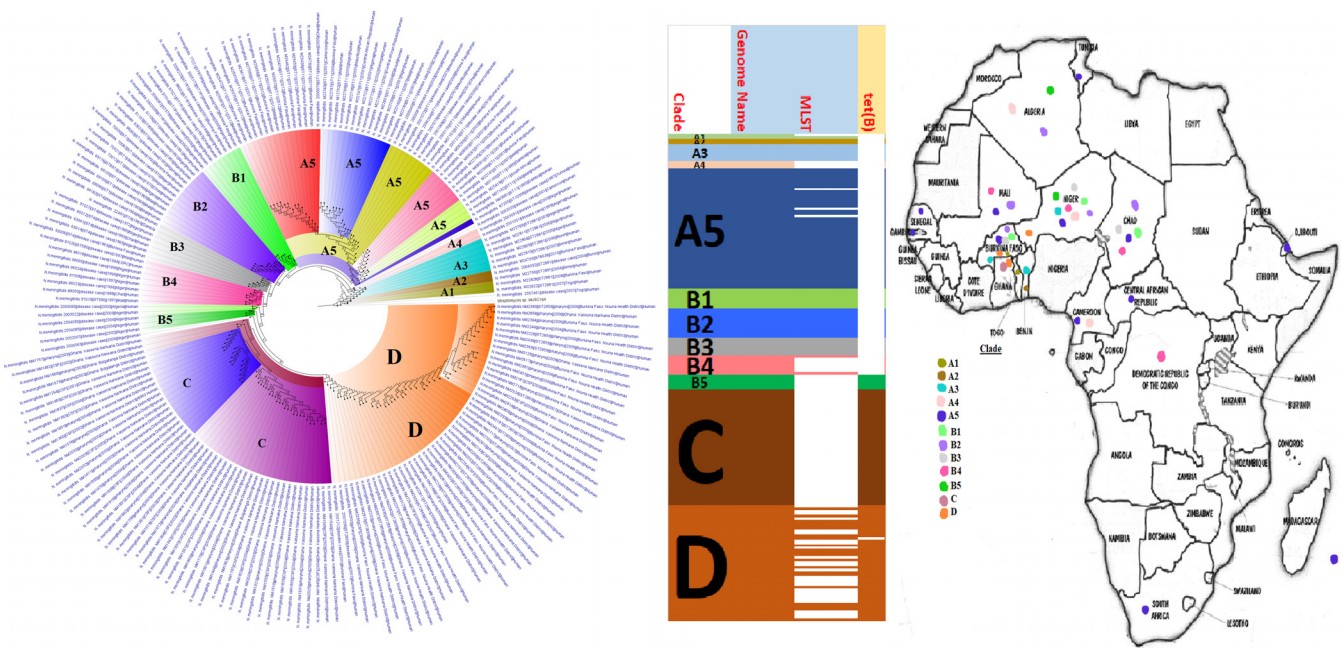

**FIG 12** Geographic distribution of *Neisseria meningitidis* clades and associated resistomes in Africa. *Neisseria meningitidis* was isolated only from human samples from South Africa, Comoros, DRC, CAR, Cameroon, West Africa, Djibouti, and Algeria; however, *N. meningitidis* was most concentrated in West Africa and the ARG [only *tet*(B)] was found only in clades B5, C, and D. Intercountry dissemination of isolates of the same clade was observed. Isolates from humans, animals, the environment, and plants are colored blue, red, mauve/pink, and green, respectively, on the phylogeny tree.

many African countries within the study period (Table S2). This is a very concerning observation given the widespread and periodic incidence of outbreaks involving this pathogen in most developing countries (18, 21).

*C. coli/jejuni* strains were reported in substantial numbers from animals, which are their natural hosts (49), as well as from human and environmental samples, albeit few genomes (all from South Africa, including *Campylobacter concisus* from human feces) of these species were available from the continent (Table 1; Fig. 16). As well, they were not as widely geographically distributed as *S. enterica* as they were reported from only Botswana (human excreta and chicken cecum), Cameroon (household water), South Africa (human excreta and river water), and Tanzania (cattle milk/beef). Moreover, they harbored relatively fewer ARGs and had generally lower resistance levels, albeit resistance to ampicillin, azithromycin, ciprofloxacin, erythromycin, nalidixic acid, and tetracycline was high (Tables S1 to S3). Interestingly, the *C. coli/jejuni* isolates were mostly multiclonal, suggesting evolutionary versatility and polyclonal dissemination. *Campylobacter* spp. are implicated in many diarrheal cases and are the major cause of human bacterial gastroenteritis worldwide, causing fatal infections in infants, the elderly, and immunocompromised patients (49). Hence, the few data available on this pathogen are disturbing as it makes it difficult to effectively plan appropriate interventions. However, their presence in humans, animals, and the environment in substantial numbers shows their host adaptability, making them ideal candidates for One Health surveillance studies.

*V. cholerae*, another common food- and waterborne diarrhea-causing pathogen implicated in recurring outbreaks in many parts of Africa (13, 50), was also reported in substantial numbers from human sources and, more importantly, from environmental sources in several countries (Table 1; Fig. 13). The higher isolation of *V. cholerae* ST69 and ST515 clones across several countries in Southern, Eastern, and Western African countries, which clustered within three main clades having very close evolutionary distance and highly conserved but rich AR gene repertoire, shows the presence of the same and highly similar strains (with little genetic diversity) circulating in Africa and causing recurring outbreaks with high morbidities and mortalities. Just like *S.* Typhi and

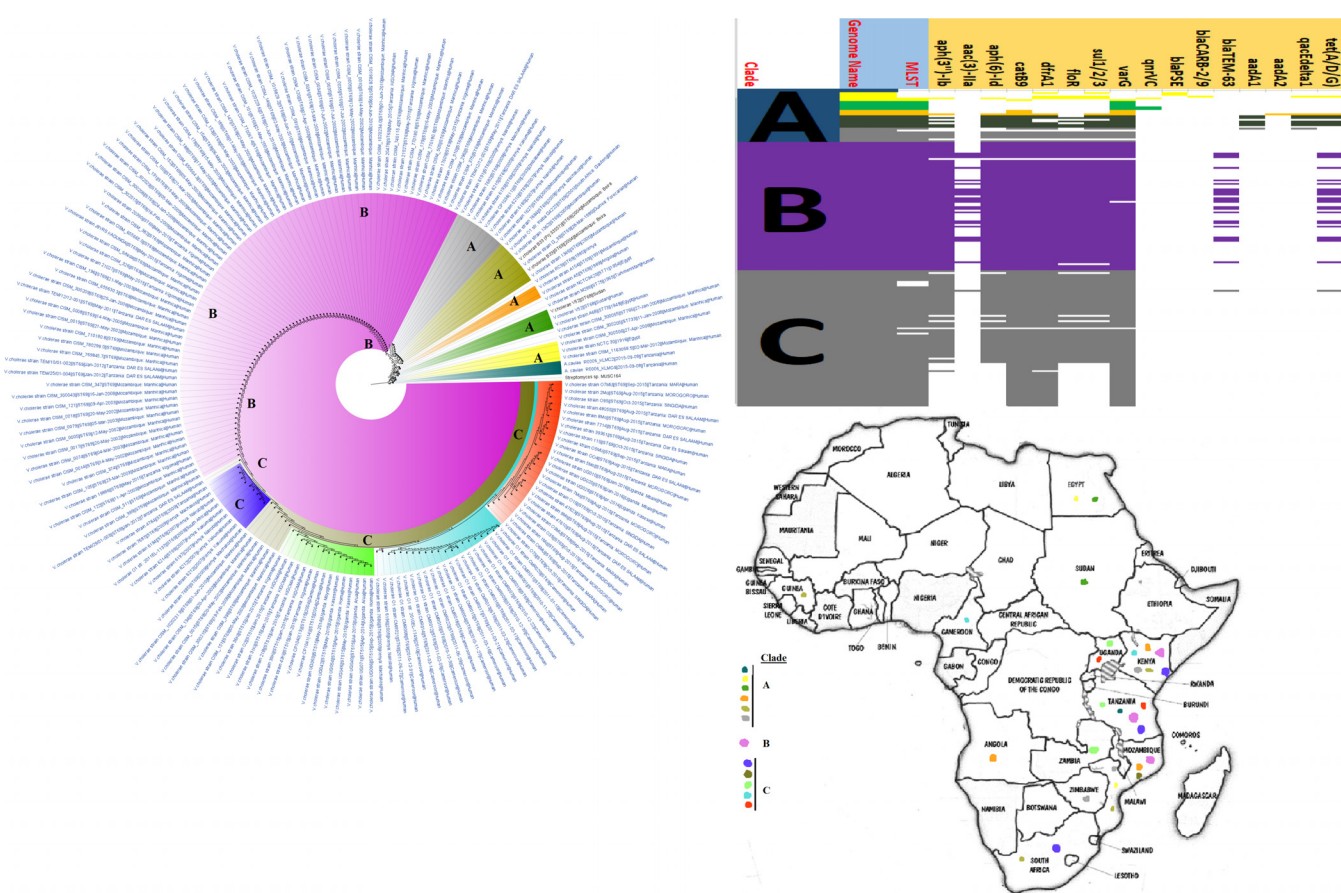

**FIG 13** Geographic distribution of *Vibrio cholerae* clades and associated resistomes in Africa. *V. cholerae* was reported only from humans in Southern and Eastern Africa, with few isolates from Cameroon, Guinea, Sudan, and Egypt. Members of clade B, which were very closely clustered together with few evolutionary variations, were mainly located in Kenya, Mozambique, and Tanzania. Clades A and C were more geographically diverse with more evolutionary variations. As supported by the resistome data, clade B had almost uniform resistomes throughout, followed closely by clades C and A, which had the smallest repertoire (collection) of resistomes. Conserved within clades B and C were *aph(3")-Ib*, *aph(6')-Id*, *catB9*, *dfrA1*, *floR*, *sul1/2/3*, and *varG*. Other ARGs in clade B were TEM-63 and *tet*(A/D/G); *aadA1/2* and *qacL/ΔE1* were found only in clade A. Intercountry dissemination of isolates of the same clade was observed. Isolates from humans, animals, the environment, and plants are colored blue, red, mauve/pink, and green, respectively, on the phylogeny tree.

*Campylobacter* spp., *V. cholerae* also causes serious diarrhea in addition to vomiting in patients and has been implicated in death in untreated patients within hours (13, 50). Subsequently, the large ARG diversity in these strains is concerning. Indeed, a carbapenemase gene termed *bla*$_{VCC-1}$, mediating resistance to carbapenems and most β-lactams, has been recently detected in *Vibrio* spp. (16, 51, 52), although this was not found in any of these isolates.

Nonfermenting Gram-negative bacilli such as *P. aeruginosa*, *A. baumannii*, *S. maltophilia*, and *Aeromonas hydrophila* are known opportunistic nosocomial pathogens with intrinsic resistance to several antibiotics (53–57). Particularly, *P. aeruginosa* and *A. baumannii*, which were two of the most common pathogens with most ARGs in Africa, are commonly implicated in several difficult-to-treat and fatal clinical infections worldwide (53–57). Thus, the higher resistome diversity, geographical distribution, isolation frequency, and resistance levels of these pathogens are not surprising. Whereas OXA-23/51-like carbapenemases are known to be common in *A. baumannii* (36, 58), the uniform presence of OXA-48-like carbapenemases in *P. aeruginosa* genomes from Africa is very worrying, particularly given the wider geographical distribution and ubiquity of this pathogen (Fig. 8) (53, 56). Owing to the broad β-lactam spectrum of carbapenemases and the importance of β-lactams in treating bacterial infections, the presence of these and other ARGs in these pathogens with high intrinsic resistance in Africa is a cause for concern (4, 36, 58). Given the difficulty in treating infections caused by

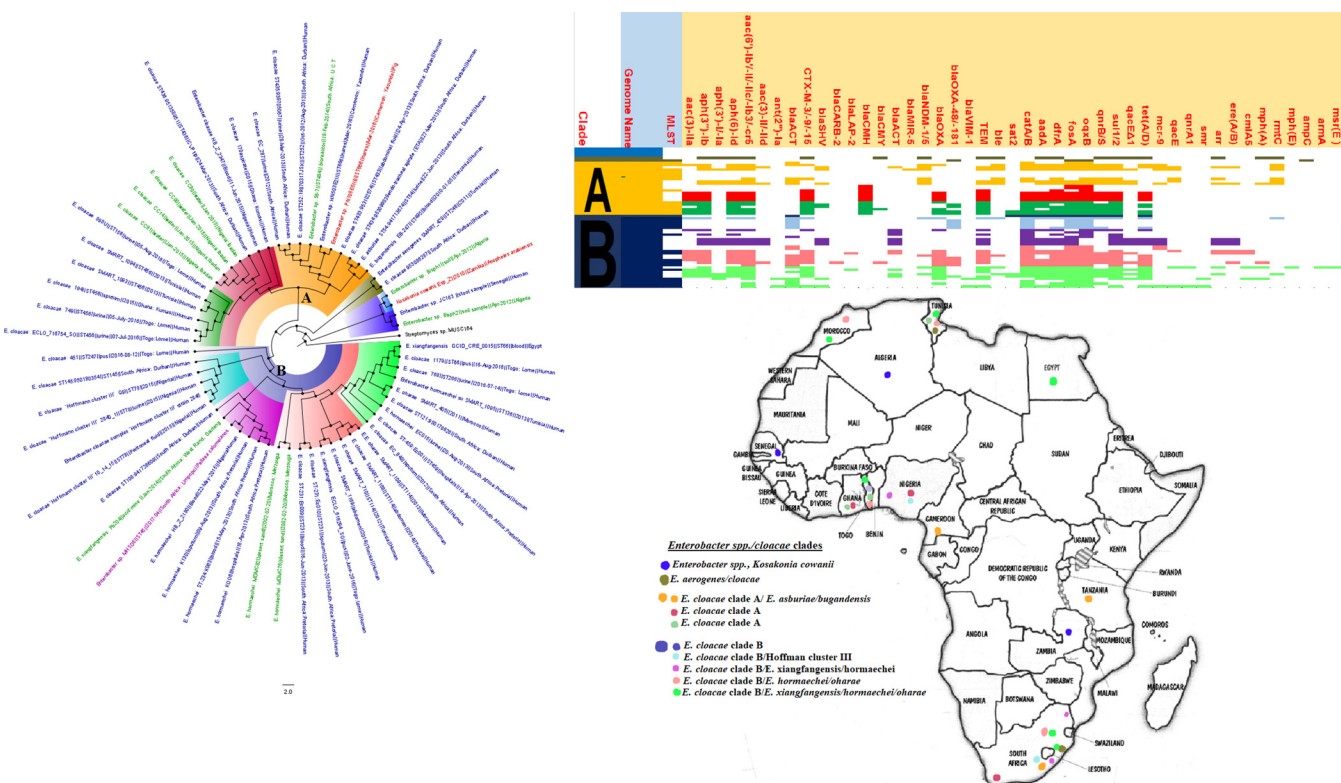

**FIG 14** Geographic distribution of *Enterobacter* clades and associated resistomes in Africa. *Enterobacter aerogenes*, *Enterobacter asburiae*, *E. cloacae*, *Enterobacter hormaechei*, and *Enterobacter xiangfangensis* were obtained from animals, humans, environment (water), and plants in mainly South Africa, Zambia, and West and North Africa. The *Enterobacter* spp. had a rich resistome repertoire (collection) ($n \geq 25$), comprising ARGs such as *aac(3')-IIa*, *aac(6')-IIa/Ib3/-IIc/Ib-cr*, *aph(3")-Ib*, *aph(3')-I/Ia*, *aph(6')-Id*, *bla*$_{ACT}$, *bla*$_{CMH}$ (clade A only), *bla*$_{NDM}$, *bla*$_{CTX-M}$, *bla*$_{OXA}$, *bla*$_{OXA-48/181}$, *bla*$_{SHV}$, *bla*$_{TEM}$, *aadA*, *catA/B*, *dfrA*, *fosA*, *oqxAB*, *sul1/2*, *qnrB/S*, *tet*(A/D), *mcr-1/9*, *arr*, and *rmtC*. Intercountry as well as human-animal-water-food dissemination of isolates of the same clade was observed. Isolates from humans, animals, the environment, and plants are respectively colored blue, red, mauve/pink, and green, respectively, on the phylogeny tree.

nonfermenting Gram-negative bacilli, it is quite refreshing to note that *S. maltophilia* and *A. hydrophila* were less often isolated with few or no ARGs.

Important sexually transmitted infections such as gonorrhea (*N. gonorrhoeae*) and nongonococcal urethritis (*M. genitalium*) and respiratory infections such as pneumonia (*M. pneumoniae*), cystic fibrosis (aggravated by *B. cepacia*), and whooping cough (*B. pertussis*), as well as cerebrospinal infections such as meningitis (*N. meningitidis*), are caused by GNB, killing millions of people annually (6–15, 59). Unfortunately, only *N. meningitidis* genomes were reported from Africa (17 countries), although both *N. meningitidis* (Egypt and Niger [serogroups C and W]) and *N. gonorrhoeae* (only Kenya) were found in the literature (Table S6; Fig. 12). Notably, *tet*(B) was the sole ARG found in *N. meningitidis* genomes, particularly clades B5, C, and D, whereas several other mechanisms (*gyrA*, *penA*, and *rpoB*) were reported in the literature (Table S2). These differences in the literature and genomic resistomes are quite interesting as some of the genomes were from Niger. Within each *N. meningitidis* clade were isolates from different countries, suggesting interboundary transmission. This is not surprising given the high transmissibility of *N. meningitidis* and the intercountry trade existing within the Sahel and West African nations with the highest concentration of this pathogen (Table S6; Fig. 12) (8).

Worryingly, two *N. meningitidis* strains from Egypt were MDR to clinically important antibiotics such as ciprofloxacin, cefotaxime, amikacin, ampicillin, penicillin, and meropenem. Sadly, reports of MDR *N. gonorrhoeae* that are highly resistant to first-line antigonococcal drugs such as third-generation cephalosporins and azithromycin are increasing, prompting revisions in treatment guidelines (6, 59–61). For instance, MDR *N. gonorrhoeae* strains that were treatable only with carbapenems have been reported in

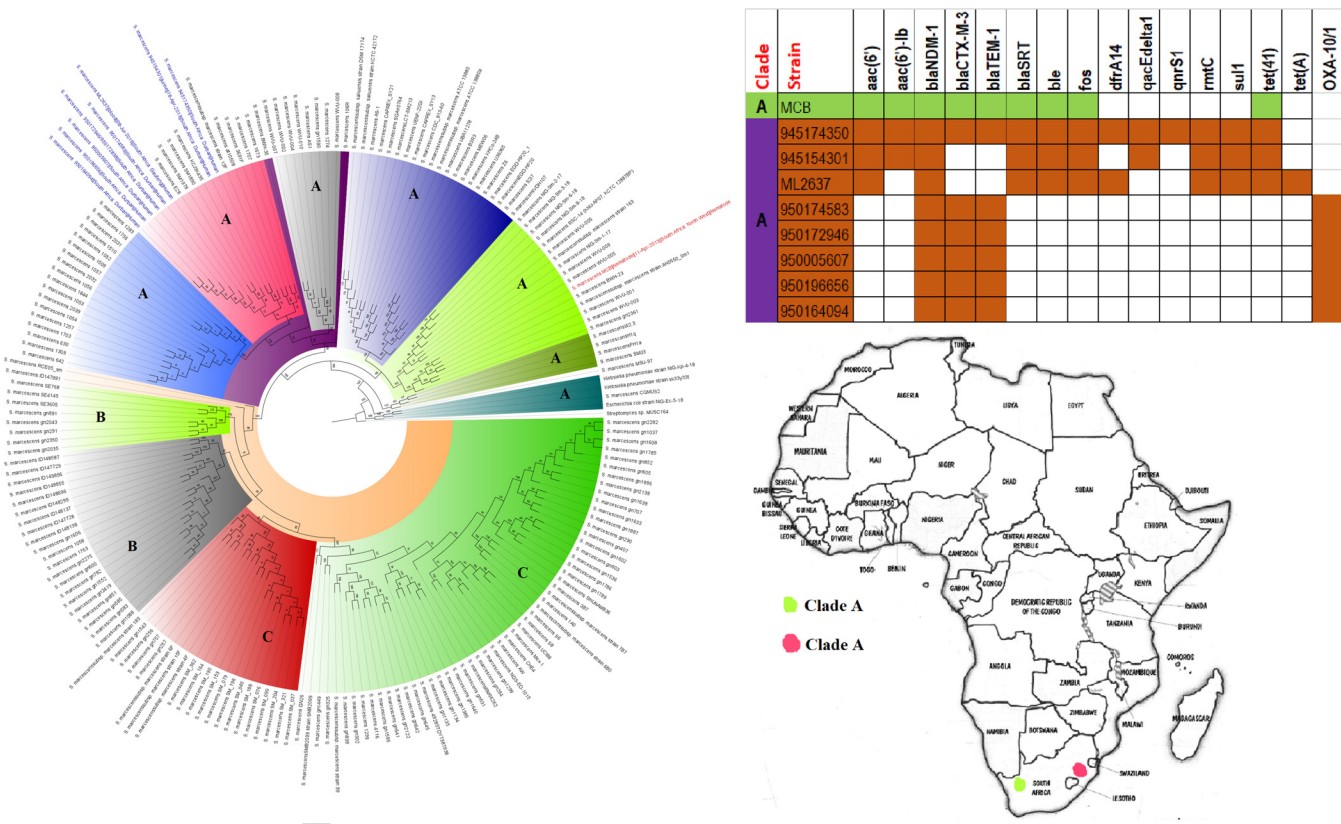

**FIG 15** Geographic distribution of *Serratia marcescens* clades and associated resistomes in Africa. The *S. marcescens* strains were mostly from humans, with one being from a nematode; they were all isolated from South Africa. NDM-1 was common in the resistomes of all the isolates, with CTX-M-3, TEM, and OXA-1 ESBLs being very common in many of the strains. Isolates from humans, animals, the environment, and plants are colored blue, red, mauve/pink, and green, respectively, on the phylogeny tree.

the United Kingdom and Australia (59–61). Unfortunately, *N. gonorrhoeae* resistance levels were not calculable due to the absence of *N. gonorrhoeae* ASTs. Indeed, the use of molecular tests to determine *N. gonorrhoeae* resistance (59), although fast, is unable to provide AST data that are critical to guide treatment. The shift to molecular tests is thus making *N. gonorrhoeae* AST data scarce and could account for the dearth of information on *N. gonorrhoeae* AST in the included articles (59).

*M. genitalium* genomes from Africa were not found, although they have been reported in South Africa as having *gyrA* and *parC* resistance determinants (Table S2). However, *M. pneumoniae* genomes, obtained from Southern, Western, Eastern, and Northern Africa, harbored no known ARGs. Other species such as *M. gallinarum*, *M. gallinaceum*, *M. pullorum*, *M. mycoides*, and *Mycoplasma capripneumoniae* were found in animals while *M. arginini* was isolated from the environment. As with *Neisseria* spp., increasing macrolide resistance in *Mycoplasma* spp. is being reported, making them less sensitive to azithromycin.

Although *B. cepacia* does not cause cystic fibrosis, it aggravates it (10). *B. cepacia* strains are inherently resistant to treatment by most antibiotics, and they normally occur alongside *P. aeruginosa* in cystic fibrotic lungs, where they can cause persistent infections and death (10, 62). In general, *Burkholderia* spp. were very few throughout Africa, in both literature and published genomes, with no known resistance gene being reported (Fig. 21). *B. pertussis*, the causative agent of whooping cough, a major childhood killer disease that has been killing infants for thousands of years (14), was rarely reported in the literature and had fewer public genomes from only Kenya and South Africa. In fact, the fewer reports of this infection in the literature do not reflect the epidemiology of whooping cough on the continent, as it kills many infants in Africa

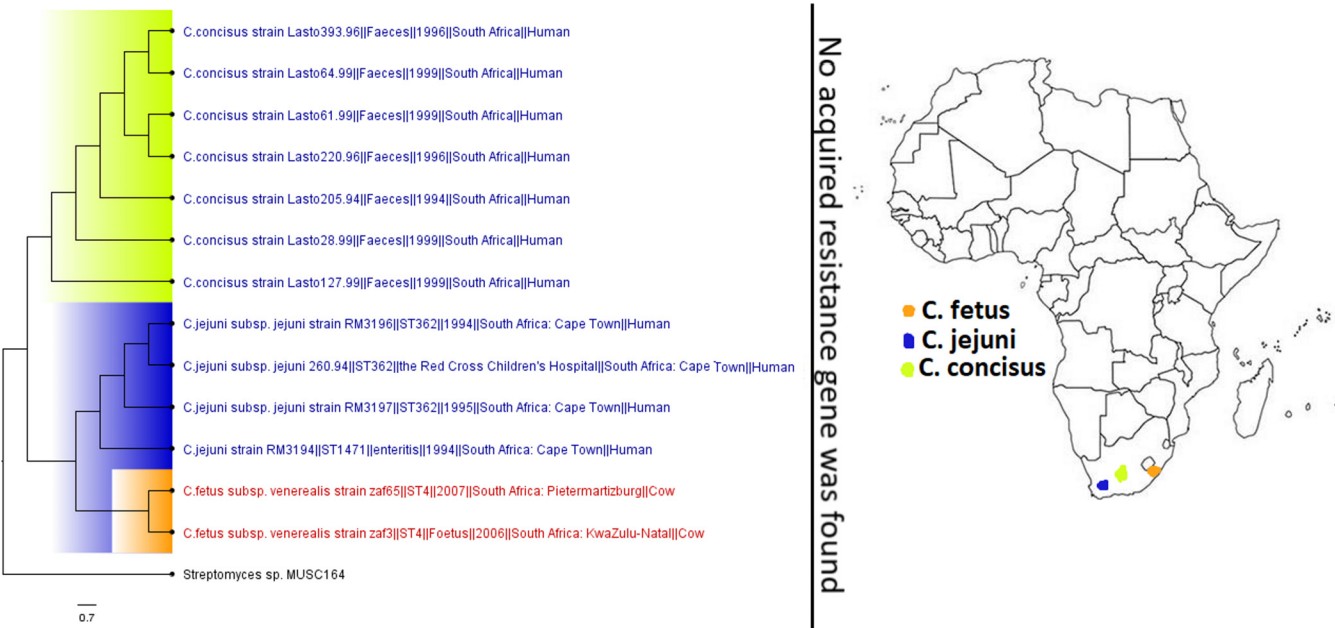

**FIG 16** Geographic distribution of *Campylobacter* clades and associated resistomes in Africa. The included genomes were all from human (*C. concisus/jejuni*) and animal (*C. fetus*) samples from South Africa with no known ARGs in the genomes. Isolates from humans, animals, the environment, and plants are colored blue, red, mauve/pink, and green, respectively, on the phylogeny tree.

yearly (63), suggesting that most clinical cases are not published. Although vaccinations have reduced the mortality and morbidity rates associated with this pathogen, variations in the pathogen are reducing the efficacy of the vaccine and increasing its reemergence globally (63).

While other *Enterobacteriaceae* (GNB) species such as *S. marcescens*, *Citrobacter* spp., *Providencia* spp., *Proteus mirabilis*, and *M. morganii* were relatively less frequently isolated than *E. coli* and *K. pneumoniae*, they did harbor a rich and diverse collection of resistomes in different clones/clades across Africa, although many of these were mainly reported from South Africa, followed by Tanzania, Nigeria, Senegal, and Egypt (Fig. 15, 17, 18, 23, and 25). These species have been implicated in fatal infections such as sepsis and wound infections (33, 36, 41, 64, 65), bearing critical ARGs such as ESBLs and carbapenemases (Table S2). On the other hand, rarely isolated/reported GNB species such as *Trabulsiella* spp., *Psychrobacter* spp., *Myroides* spp., *Chromobacterium* spp., *Brevundimonas* spp., and *Bacillus* spp. were mainly obtained from the environment or termites in very few countries and harbored no ARGs. As well, *A. faecalis*, *Leclercia adecarboxylata*, *Pantoea* spp., and *Raoultella* spp., which were mainly isolated from human samples, except for *Pantoea* spp., which were also obtained from animals and humans, also contained no known ARGs and were mainly restricted to a few geographical areas, except *Pantoea* spp., which were widely distributed across Africa (Table 1 and Tables S1 to S3; Fig. 20 and 24 and Fig. S3).

The included articles did not undertake a One Health study but focused on a single ecological niche, *viz.*, animal, human, and environmental samples. Thus, the epidemiological relationship between these isolates from these individual studies was instead shown by the phylogenetic trees drawn from the individual genomes in this study. The close evolutionary relationships observed between these genomes from different ecological niches and countries proffer a stronger support for One Health studies to facilitate easy epidemiological analyses of infectious diseases in Africa and globally.

**Future perspectives and conclusion.** The backbone to efficient diagnosis and treatment of infections is rapid, effective, simple, and inexpensive diagnostics and skilled laboratory scientists (66–68). Without appropriate diagnostics, the etiology of many infectious diseases, the genotype/clone of the infecting organism, and its resis-

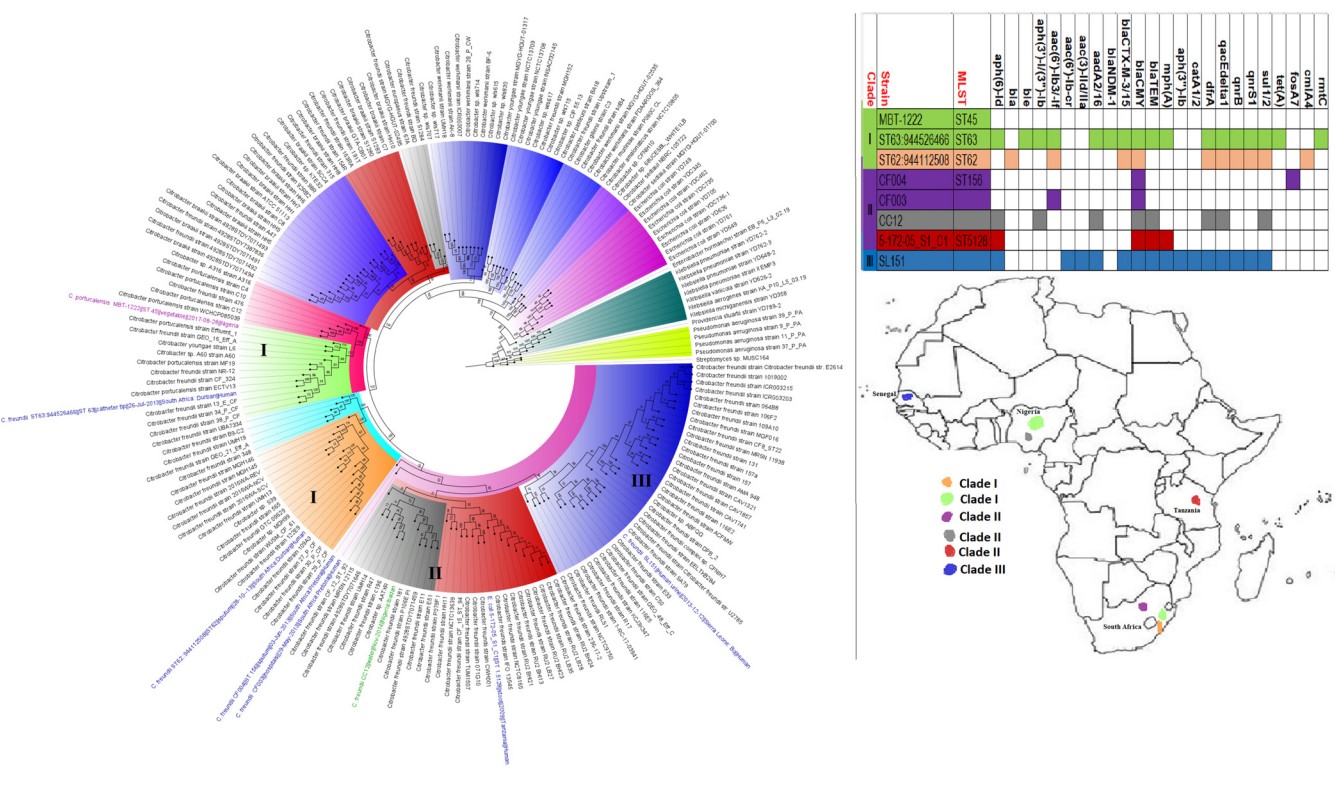

**FIG 17** Geographic distribution of *Citrobacter* clades and associated resistomes in Africa. The *Citrobacter* genomes were from humans and water in South Africa, Tanzania, Nigeria, and Senegal. $bla_{CMY}$ was the only ARG conserved in all the clades, while clade I had more ARGs; generally, the ARGs were not clade specific but isolate specific. Isolates from humans, animals, the environment, and plants are colored blue, red, mauve/pink, and green, respectively, on the phylogeny tree.

tance mechanisms cannot be known, and a proper therapeutic choice cannot be made (66–68). This describes the situation in many African countries, making several preventable infections emerge subtly into large-scale outbreaks (67, 68). As shown in Table S7, simpler phenotypic diagnostic tests with a longer turnaround time of at least 24 h were more commonly used in Africa while complex, skill-requiring, and expensive tests like whole-genome sequencing and Vitek were hardly used. These challenges affect the fight against infectious diseases and make surveillance studies on the continent difficult (66, 69).

It is worthy of consideration that as large as the number of articles and genomes used in this study is, the dearth of genome sequencing and molecular ARG surveillance in many African countries, influenced by low funding, absence of molecular diagnostic laboratories, and inadequate skilled personnel, affects the comprehensiveness of the geographic location and resistome evolutionary epidemiology (4). This is particularly true for certain species such as *V. cholerae*, *C. coli/jejuni*, *N. meningitidis/gonorrhoeae*, *Mycoplasma genitalium*, *Providencia* spp., *M. morganii*, *C. freundii*, *P. mirabilis*, *B. cepacia*, and *B. pertussis*, which are very important clinical pathogens implicated in substantial morbidities and mortalities (19). The diversity and rich abundance of ARGs described in the various species could also be affected by their relative sample sizes as species with larger genomes are more likely to have richer ARGs, although this was always not the case as seen with species such as those of *Pantoea* and *Bacillus*. Furthermore, the absence of certain ARGs and species in certain countries and/or ecological niches does not necessarily suggest their total absence thereat but could be due to the fact that the included studies and genomes might not have focused on them or used diagnostics that could have identified them.

In summary, MDR GNB clinical pathogens implicated in high morbidities and mortalities are circulating in Africa in single and multiple clones, shuttling diverse

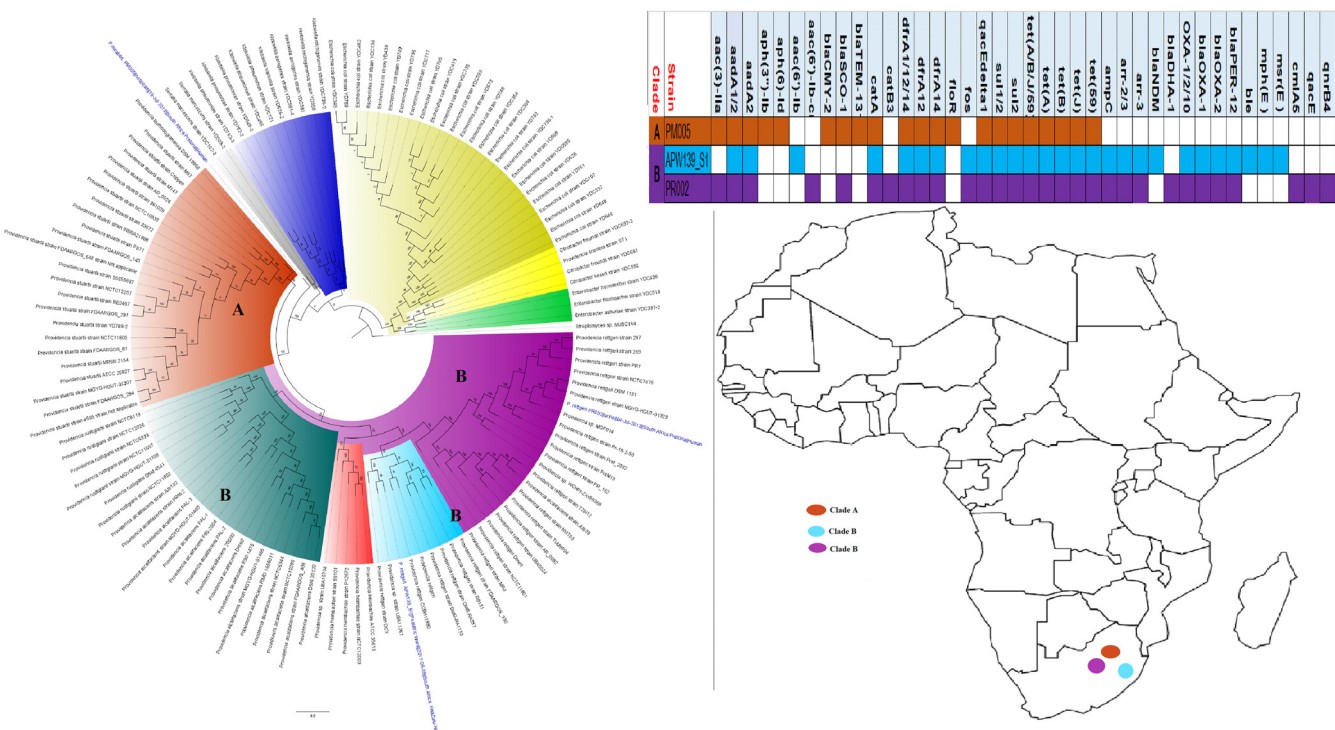

**FIG 18** Geographic distribution of *Providencia rettgeri* clades and associated resistomes in Africa. All the *P. rettgeri* strains were from South Africa and contained several ARGs. Isolates from humans, animals, the environment, and plants are colored blue, red, mauve/pink, and green, respectively, on the phylogeny tree.

resistomes on plasmids, integrons, insertion sequences, and transposons from animals, foods, plants, and the environment to humans. A comprehensive One Health molecular surveillance is needed to map the transmission routes and understand the resistance mechanisms of these pathogens to inform appropriate epidemiological interventions.

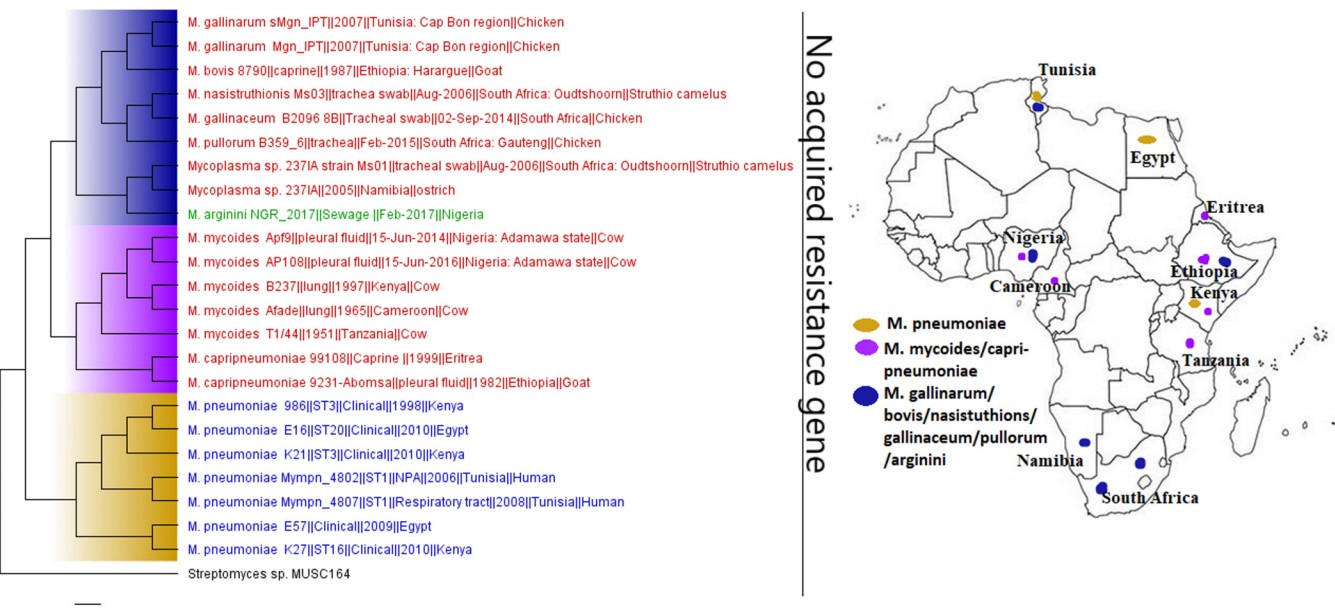

**FIG 19** Geographic distribution of *Mycoplasma* clades and associated resistomes in Africa. These strains were host specific, with *M. gallinarum/pullorum/gallinaceum/mycoides/capripneumoniae* being found in animals and *M. pneumoniae* being found in humans; *M. arginini* was found in the environment. These strains were mainly found in Egypt, Cameroon, Nigeria, Tunisia, and Southern and Eastern Africa. No known ARGs could be found in the genomes. Isolates from humans, animals, the environment, and plants are colored blue, red, mauve/pink, and green, respectively, on the phylogeny tree.

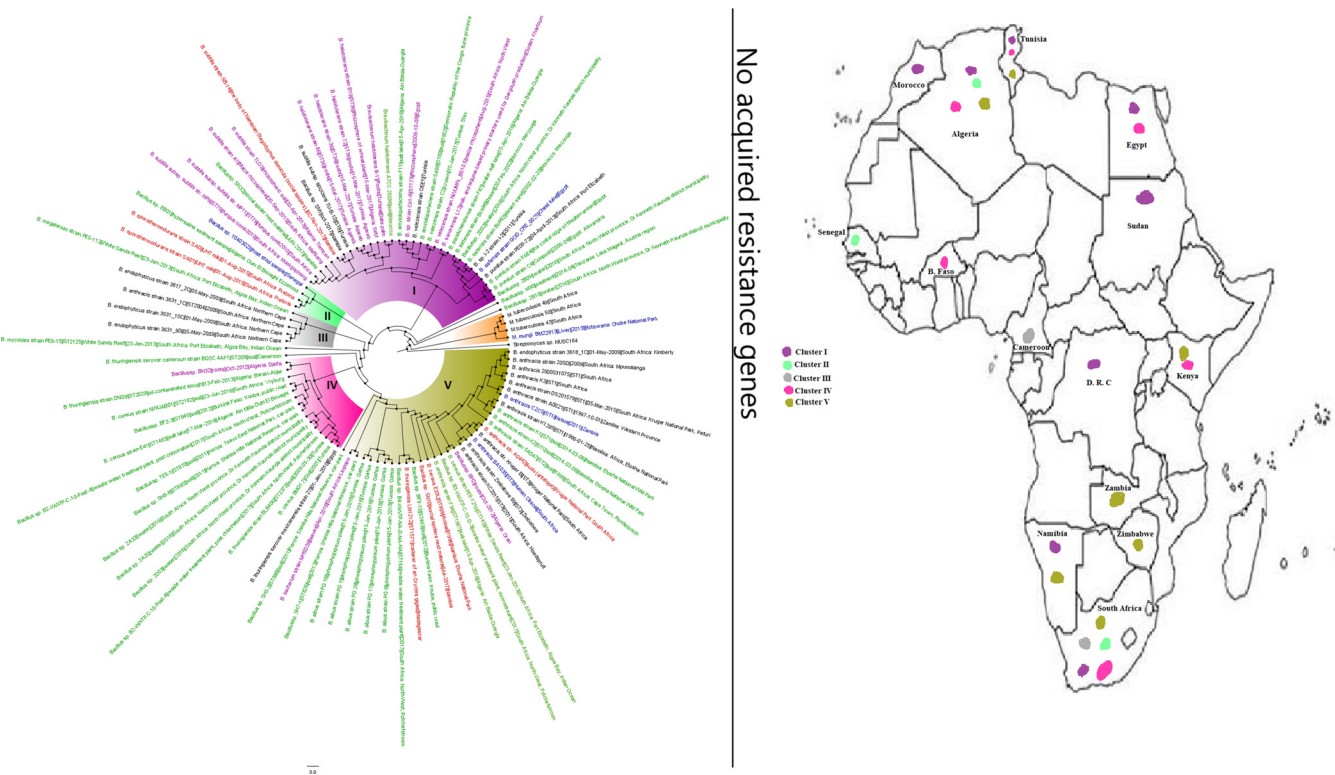

**FIG 20** Geographic distribution of *Bacillus* clades and associated resistomes in Africa. There were five main clusters, comprising various species of this genus, which were from animals (e.g., *Bacillus anthracis*, *Bacillus cereus*, *Bacillus thuringiensis*), humans (*B. anthracis*), environment (e.g., *B. anthracis*, *B. cereus*, and *B. thuringiensis*), and plants (e.g., *B. subtilis*, *Bacillus halotolerans*, and *Bacillus velezensis*) in Southern Africa, DRC, Kenya, Cameroon, B. Faso, Senegal, and North Africa. No known ARGs could be found in the genomes. Isolates from humans, animals, the environment, and plants are colored blue, red, mauve/pink, and green, respectively, on the phylogeny tree.

The comprehensive genomic and extracted literature data presented in this work provide an important foundation for future studies on GNB epidemiology in Africa. Future works must therefore investigate a meta-analysis of the data presented here.

## MATERIALS AND METHODS

**Databases and search strategy.** A comprehensive literature search was carried out on PubMed and/or ResearchGate, ScienceDirect, and Web of Science electronic databases. Research articles published in the English language between January 2015 and December 2019 were retrieved and screened using the following search terms and/or phrases: "molecular epidemiology," "gram-negative bacteria," "mechanisms of resistance," "antimicrobial resistance genotypes," "drug resistance," "AMR genotypes," "genetic diversity," "clones," "genotyping," "antibiotic resistance gene," "plasmid," "mobile genetic elements," "resistome," "gene mutation," "resistance gene mutation," and "Africa." Each search term was paired with every other search term in addition to the term "Africa"; this was repeated by replacing "Africa" with each African country in a factorial fashion. The search terms were separated by the "AND" Boolean operator. The "OR" Boolean operator was used only between "mechanisms of resistance" and "drug resistance" and "antibiotic resistance gene."

Articles published within January 2015 to December 2019 were included in this review to provide a current quinquennial epidemiology of GNB and their AR gene dynamics in Africa. Data mining, title and abstract screening, and data extraction were undertaken by both authors independently, after which the results were cross-checked and conflicting outcomes were resolved by both authors, based on the inclusion criteria and filters.

**Inclusion and exclusion criteria.** Articles addressing the molecular mechanisms (using PCR, microarray, or whole-genome sequencing [WGS]) of AR in GNB and undertaking bacterial typing (MLST, PFGE, ERIC-PCR, and WGS) were included in this systematic review. Papers that addressed only phenotypic resistance were excluded (Fig. 1A). Studies that did not include GNB isolates from Africa and were not written in English within 2015 to 2019 were excluded. Emphasis was placed on year of publication and not year of isolation or investigation; hence, studies conducted in 2010 but published within 2015 to 2019 were included. Included under animals are livestock, pets, wildlife, and animal food products such as milk, meat, and eggs while plants and plant foods were subcategorized under the environment.

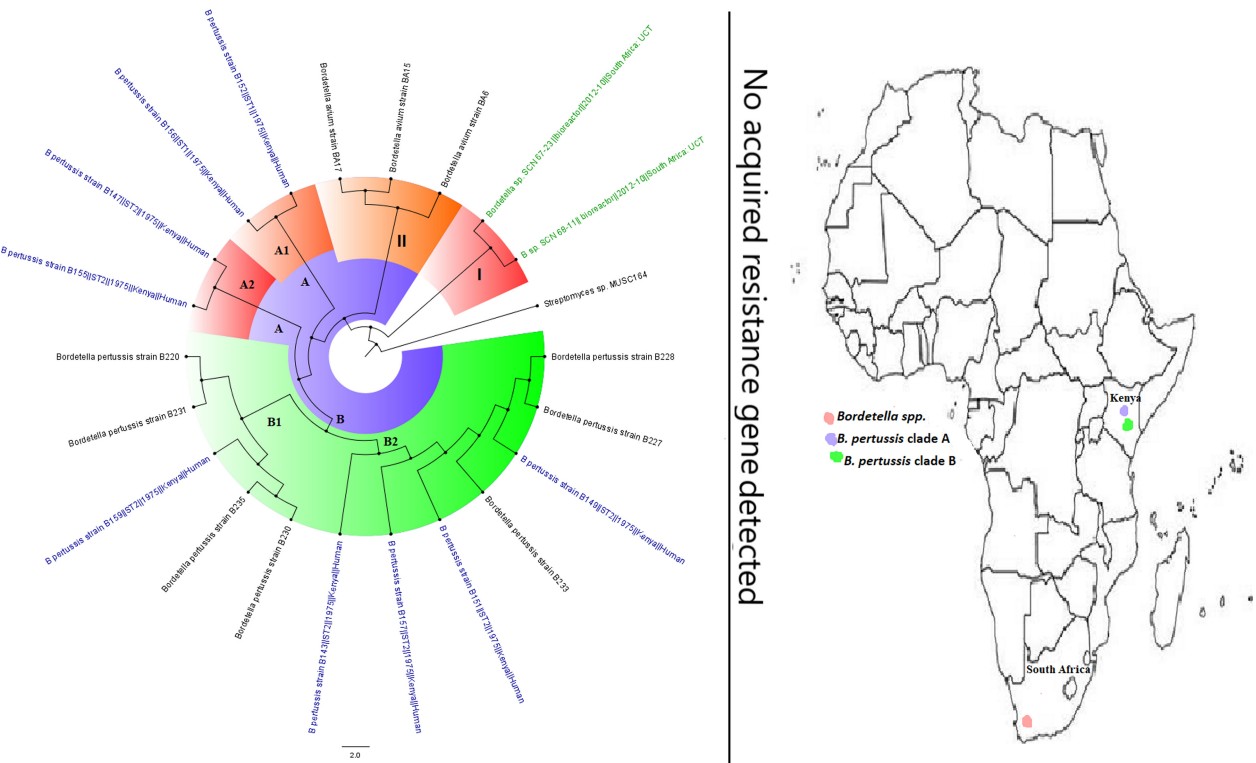

**FIG 21** Geographic distribution of *Bordetella* clades and associated resistomes in Africa. The genomes were obtained from only Kenya from humans (*B. pertussis*) or the environment (other species). No known ARGs could be found in the genomes. Isolates from humans, animals, the environment, and plants are colored blue, red, mauve/pink, and green, respectively, on the phylogeny tree.

**Included data.** The following data were extracted from the included articles: country, study year, sample type and source(s), sample size, total isolates (isolation rate), total isolates for which antimicrobial sensitivity testing (AST) was performed, bacterial species, clone/MLST (multilocus sequence typing), antibiotic resistance genes (ARGs), mobile genetic elements (MGEs), antibiotic resistance phenotype, genotyping, and diagnostic method(s)/techniques used (see Tables S1 to S3 in the supplemental material).

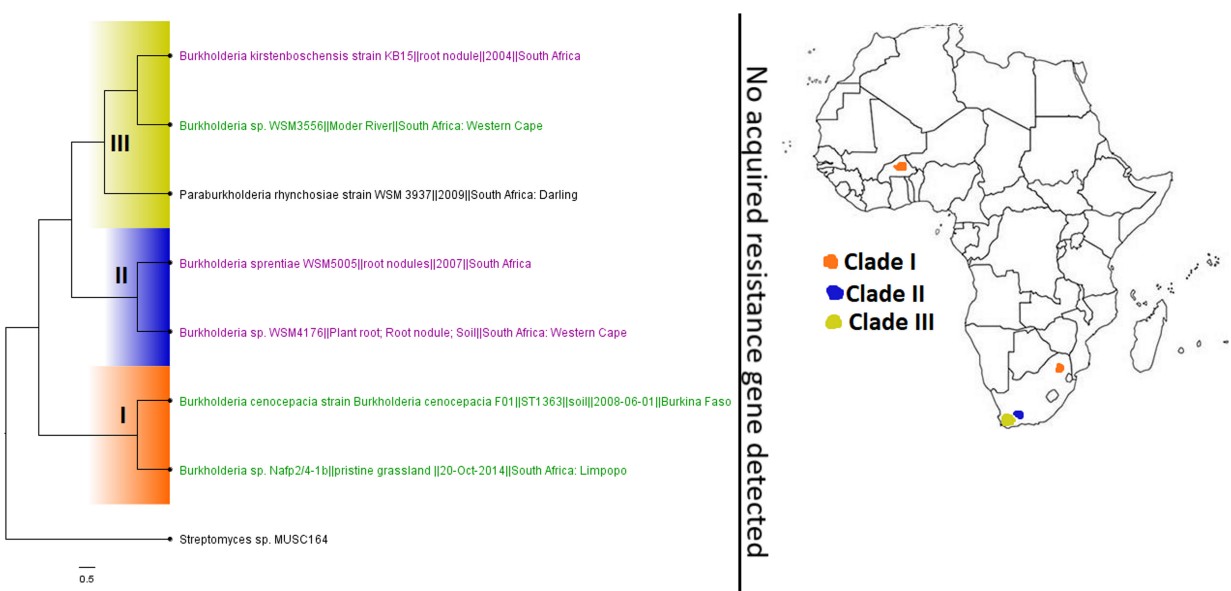

**FIG 22** Geographic distribution of *Burkholderia* clades and associated resistomes in Africa. *Burkholderia* genomes in Africa were solely from South Africa and B. Faso in plants and environmental samples. *B. cepacia* genomes were not found in Africa, and no known ARGs could be found in the included genomes. Isolates from humans, animals, the environment, and plants are colored blue, red, mauve/pink, and green, respectively, on the phylogeny tree.

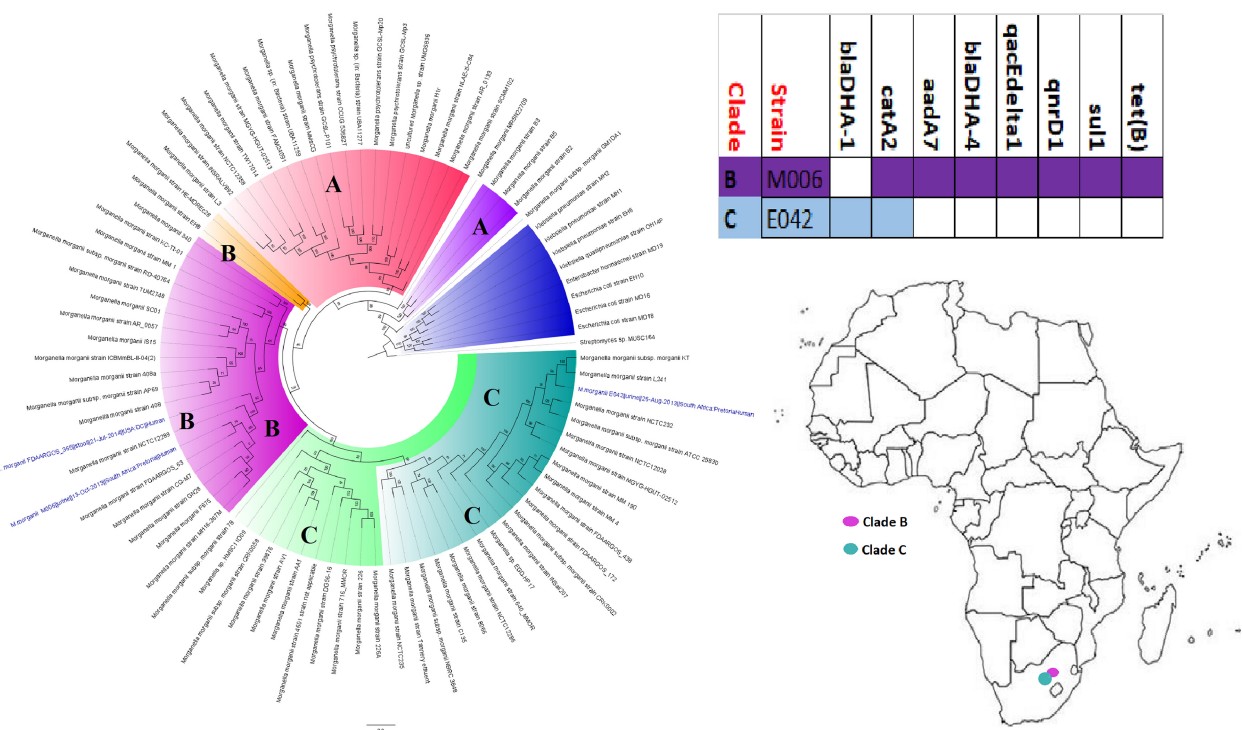

**FIG 23** Geographic distribution of *Morganella morganii* clades and associated resistomes in Africa. Only two isolates from humans in South Africa were obtained for all of Africa; one of these two strains had a relatively rich resistome. Isolates from humans, animals, the environment, and plants are colored blue, red, mauve/pink, and green, respectively, on the phylogeny tree.

**Relative ratio and bioinformatic analyses.** Microsoft Excel 365 was used to analyze the frequencies using raw data extracted from all included articles. For each GNB species, the resistance levels per antibiotic per country were calculated by dividing the total resistant isolates by the total isolates for which antibiotic sensitivity was determined. Frequencies of resistant species, clones, ARG-MGE associations, and AR levels were evaluated per animal, human, and environmental source per country (Tables S1 to S3). In calculating the levels of resistance of each species to an antibiotic per country, we used absolute cutoffs of either resistant or susceptible, i.e., the isolate was defined as

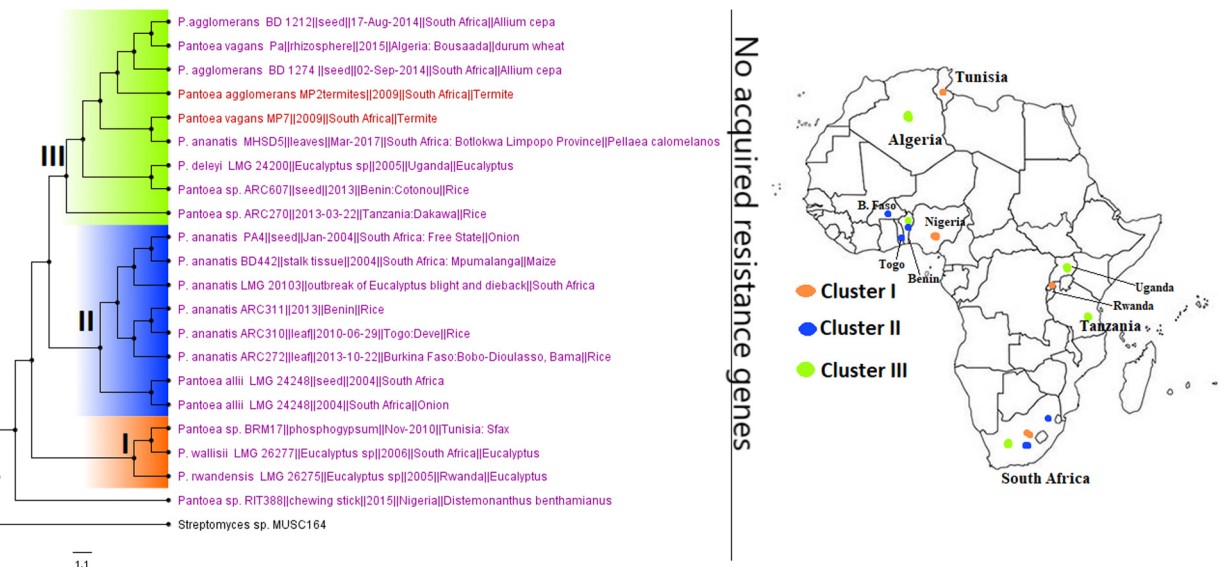

**FIG 24** Geographic distribution of *Pantoea* clades and associated resistomes in Africa. *Pantoea* spp. were mainly found in the environment and in termites in South Africa, Tanzania, Rwanda, Uganda, West Africa, Algeria, and Tunisia. No known ARGs could be found in the genomes. Intercountry as well as plant-animal (termite) dissemination of isolates of the same clade was observed. Isolates from humans, animals, the environment, and plants are colored blue, red, mauve/pink, and green, respectively, on the phylogeny tree.

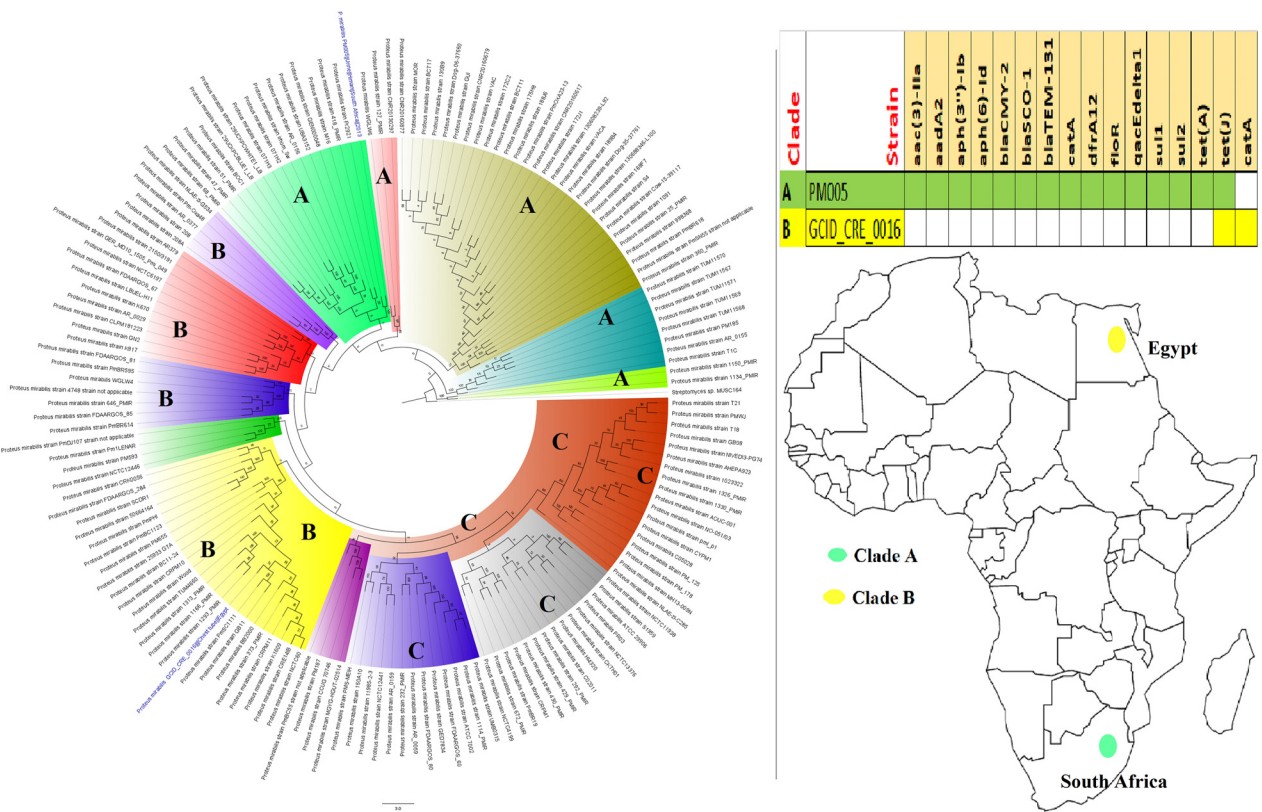

**FIG 25** Geographic distribution of *Proteus mirabilis* clades and associated resistomes in Africa. *P. mirabilis* genomes were found only in human samples from Egypt and South Africa, with substantial differences in their resistomes. Isolates from humans, animals, the environment, and plants are colored blue, red, mauve/pink, and green, respectively, on the phylogeny tree.

either resistant or susceptible to the antibiotic in question. The type of diagnostic or breakpoint benchmark, *viz.*, CLSI or EUCAST, used to measure and define the AR in the isolates was not considered in this reckoning. Thus, if the authors defined the isolate as resistant, it was used as such in calculating the resistance levels. Countries with AR levels above 50% for an antibiotic were defined as having high resistance to that antibiotic while those with AR levels below 50% were defined as low. AR levels above 50% were colored red for each antibiotic in Tables S1 to S3 while those below 50% were colored black.

Analyses (counts and relative ratios) of species distribution per country and across countries, species distribution per ecological niche/source (animal, humans, and environment) within countries, ARG distribution per ecological niche/source within and across countries, and resistome diversity among species were undertaken using Microsoft Excel.

Genomes (*n* = 3,028) of 24 genera that were found in the included articles and isolated from Africa were downloaded from Pathosystems Resource Integration Center (PATRIC) (https://www.patricbrc.org). Genomes of GNB which were not reported in the included articles and that had no genomes on PATRIC were not included. The genomes of each species were aligned using the multiple sequence alignment tool in PATRIC (https://docs.patricbrc.org/tutorial/alignments/multiple_sequence_alignment.html). Among the aligned genomes of each species, those that did not share at least a core of 1,000 proteins with all the aligned genomes were removed; such genomes were excluded to make the genomes "treeable." For each species, a maximum of 200 genomes, including the reference outgroup genome from *Streptococcus mitis*, were organized into batches for the phylogenetic analyses.

Species with fewer than 200 genomes were organized into a single tree while those with more than 200 genomes were divided into batches of 200 to increase the resolution of the final trees and enable easy analyses by the phylogenetic algorithm. The aligned sequences (of ≤200 genomes) were subsequently used for phylogenetic analyses using Randomized Axelerated Maximum Likelihood (RAxML)'s maximum-likelihood (version 8.2.11) method on PATRIC and annotated with Figtree (http://tree.bio.ed.ac.uk/software/figtree/) (Table S4); default parameters (PThreads version; maximum allowed deletions, 3; maximum allowed duplications, 3; GTRCAT model used) were used to run the phylogenetic reconstruction with 1,000× bootstrap resampling analyses. The AR genes of these genomes were curated from the Isolates Browser database of NCBI (https://www.ncbi.nlm.nih.gov/pathogens/isolates#/search/#/search/) by using the genomes' accession or biosample number to retrieve their ARGs (Tables S5 and S6). The geographic location (country of isolation) of the resistant clones/clades per species was mapped manually onto an African map to show their geographical distribution in Africa using colored circles representative of the color of the clades on the phylogenetic trees (Fig. 2 to 26 and Fig. S1 to S3).

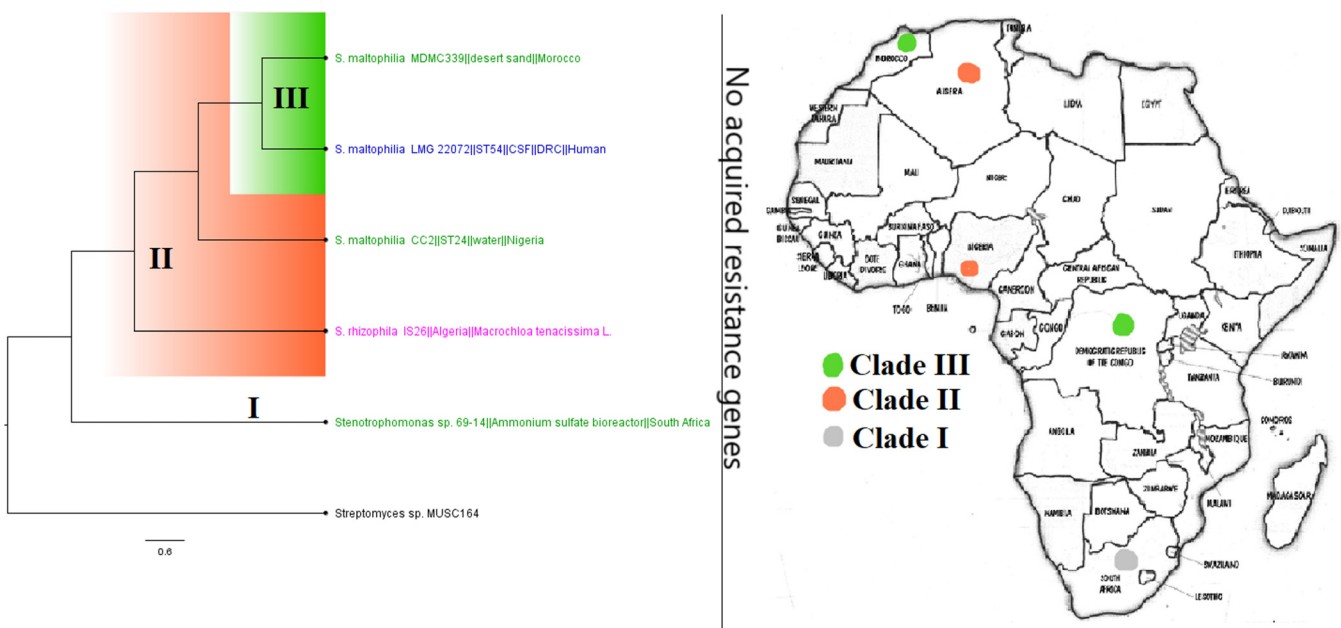

**FIG 26** Geographic distribution of *Stenotrophomonas maltophilia/rhizophila* clades and associated resistomes in Africa. These two species were found in humans, water/environment, and plants (*S. rhizophila* only) in South Africa, DRC, Nigeria, Algeria, and Morocco, with no known ARGs in the genomes. Isolates from humans, animals, the environment, and plants are colored blue, red, mauve/pink, and green, respectively, on the phylogeny tree.

**Data availability.** All data used in this article are contained here in the supplemental material.

## SUPPLEMENTAL MATERIAL

Supplemental material is available online only.

**FIG S1**, TIF file, 2.9 MB.
**FIG S2**, TIF file, 1.9 MB.
**FIG S3**, TIF file, 2.1 MB.
**TABLE S1**, XLSX file, 0.1 MB.
**TABLE S2**, XLSX file, 0.4 MB.
**TABLE S3**, XLSX file, 0.1 MB.
**TABLE S4**, XLSX file, 1.2 MB.
**TABLE S5**, XLSX file, 0.3 MB.
**TABLE S6**, XLSX file, 2 MB.
**TABLE S7**, XLSX file, 0.4 MB.

## ACKNOWLEDGMENTS

We are exceptionally grateful to Dora Osei Sekyere of the University of Education Winneba, Kumasi Campus, Ghana, for aiding in the curation, annotation, and design of the genomic and phylogenomic metadata and trees.

There was no funding.

J.O.S. conceived and designed the study; searched the literature; analyzed the data; undertook all bioinformatic analyses, image designs, and tabulations; and wrote the paper. M.A.R. searched the literature, extracted the data and resistance genes from NCBI into Excel sheets, and undertook count analyses of the raw data.

We declare no conflict of interest.

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
