## [Reviewer comments · mSystems]

Genomic and Resistance Epidemiology of Gram-Negative Bacteria in Africa: A Systematic Review and Phylogenomic analyses from a One-Health Perspective

John Osei Sekyere and Melese Reta

Corresponding Author(s): John Osei Sekyere, University of Pretoria

Review Timeline:

Submission Date:	September 8, 2020
Editorial Decision:	October 15, 2020
Revision Received:	October 19, 2020
Accepted:	October 28, 2020

Editor: Zarath Summers

Reviewer(s): The reviewers have opted to remain anonymous.

Transaction Report:

DOI: <https://doi.org/10.1128/mSystems.00897-20>

mSYSTEMS	
Reviewer #1 (Comments for the Author):	AUTHOR RESPONSES
Summary	
The topic of this review is pertinent and interesting; and the manuscript is generally well written. Although there have been some reviews on individual Gram-negative species in Africa, there is no recent review that have considered most of them. The authors did a good literature search and put their findings in the proper context. However, there are some issues that prevents its publication in its current state, see major comments.	Thanks for the compliments
Major comments	
1.- Some information need to be added to the manuscript. First, in the introduction the authors want to briefly describe One Health; so that, readers not familiar with that term can fully appreciate its implications. Secondly, the authors need to explicitly state how the alignments, on which the ML phylogenies were run, were constructed; I guess these were based on core genome regions. Third, considering the MLST variable in some supplementary tables, I assume that the number after the dot is the ST, right? Furthermore, I imagine that for A. baumannii , where you have entries like "MLST.Acinetobacter_baumannii_1.231, MLST.Acinetobacter_baumannii_2.1", the first part refers to Oxford MLST scheme and its sequence type, which in this case is ST231; whereas the second part refers to ST1 from the Pasteur MLT scheme, correct? Anyhow, the authors should add a note explaining how to read MLST and other variables in the supplementary tables.	Thanks for the deeper insight. Further explanation has been in lines 74-78. The genomes were aligned with an alignment tool on PATRIC to obtain at least 1000 core shared proteins (line 143-155). The aligned files were then used to run the RxML phylogenetics. Thanks a lot or clarifying the MLST scheme. This has been explained in the legend of Table S4 in lines 908-910.
2.- As far as the antibiotic resistance mechanisms from the analyzed studies, are they really sure about the presence/absence of some antibiotic resistance mechanisms? To what extent there was actual absence of a particular resistance mechanism or rather that mechanism was not tested. For instance, the authors note that fluoroquinolone resistance mechanisms were not found in environmental isolates but are they sure that the original studies actually conducted PCR testing of the genes conferring fluoroquinolone resistance and/or antimicrobial susceptibility tests for fluoroquinolones, in which case is a true negative; or maybe, the original studies did not conduct antimicrobial susceptibility tests nor PCR testing of genes conferring fluoroquinolone resistance and, thus, it is uncertain if the environmental isolates are resistant to fluoroquinolones.	We agree with the reviewer on this point that the absence of the ARGs are limited to what was found in the studies. This has been clarified in line 287 and 568-575
3.- The visual presentation of the data can be considerably improved, not least the ML trees. Many a time the labels are way too long. Why did the authors choose cladograms instead of additive trees? Those trees do not allow to have an idea of the amount of evolutionary change (the actual branch lengths). This is just a suggestion but rectangular tree layouts maybe better, given that some of the trees have quite a few isolates. In some Supplementary figures, for instance supplementary figure 6, the x-	We agree with the reviewer on this point. The trees were made in low resolution to reduce their sizes for peer review, but a higher resolution image is now uploaded for production. We have also changed the trees to additive to show the evolutionary trajectory of the isolates. Not all trees could be done

labels are upside down. Finally, the resolution of some figures is rather poor.	this way as doing so made them less visible/awkward. Rectangular trees were used for trees with less genomes
4.- The title is rather convoluted and, given that mapping the geographic location of isolates on a phylogeny is a bit shy of a proper phylogeographic study, I would suggest this simplified version of the title:	Thank you very much for this suggestion. The title has been changed as suggested.
Resistome of Gram-Negative Bacteria in Africa: A Systematic Review and Genomic Meta-Analysis from a One-Health Perspective [1] [2] [SEP]	Title has been changed to "Genomic and Resistome epidemiology of Gram-Negative Bacteria in Africa: A Systematic Review and Phylogenomic Analysis from a One-Health Perspective"
Minor comments	
Lines 15-18, state that you are referring to Africa.	"In Africa" has been added in line 18
Line 20 and also in the discussion, would not it better "sentinel species" rather than "model/index organisms"	Thank you. These have been changed as suggested.
Line 23, a more natural acronym is just "AR" - there and throughout the manuscript	Thank you. This has been changed throughout the text
Lines 30-31 and line 120, "the geographic location" instead of "phylogeography"	Phylogeography has been changed to geographic location throughout the article
Line 107, "For each GNB species" rather than "To each GNB species"	Thank you. This has been changed.
Lines 143-144, please revise figures for the isolation rate of environmental specimens; does not make much sense.	The figures for the environmental specimens are correct. This is because several isolates were obtained from fewer samples. For instance, several isolates (~10-30) can be isolated from a single environmental sample and this is correct in practise.
Lines 147-153 and lines 325-336, those paragraphs may be better in methods.	The methods used in this article are the steps involved in gathering the data and analysing them, whilst the results refer to what was obtained from the data and its analyses. Hence, we are convinced that these lines belong to the results and not methods.
Line 161, Neisseria species do not seem to be very common as per Table 1.	"common" has been deleted from the sentence in line 156
Line 231, should be "rather than"	This has been corrected in line 269

Line 303, N. meningitis must be in italics	Italicized in line 341
Line 377, you mean little clonal dissemination of one or two prevalent lineages, right?	Yes. The sentence has been qualified as such in line 416
Line 422, "with little genetic diversity" sounds better.	Thank you. This has been changed in 459
Line 467, correct resistance	Thanks. Corrected.
Reviewer #3 (Comments for the Author):	
The goal of this study was to provide a comprehensive survey of antibiotic resistance gene-carrying pathogens across several African countries. The authors conducted a meta-analysis of English-language articles on infectious disease articles from 2015-2019, extracting information on the diagnosis, identity, and resistance elements present in these infections. This is novel, fits well within the scope of mSystems, and will be of great interest to epidemiologists, clinicians, and infectious disease experts.	Thanks for the complement, we deeply appreciate it.
The authors clearly and succinctly lay out the scope of the problem, and the gap in knowledge that their study fills. The study is impressive and ambitious in scope, and has the potential to be a critical source of information for infectious disease epidemiology in Africa. However, there is an absence of statistical analyses in this manuscript, which makes the authors' conclusions difficult to interpret. I think the data is well set up for such analyses; I have suggested a few below. Overall, I think this is an important study that needs further analyses to make it more impactful. There are some very minor grammatical issues, but these do not interfere with the messages in the paper.	Thanks for the complement, we deeply appreciate it.
Major comments:	
 The database search strategy is somewhat unclear (lines 80-89). The authors mention several search terms and that they were paired and combined, but not the precise methodology of these combinations. Was a full factorial design used (e.g. each search term paired with each other search term)? Was the search term "Africa" always included? Were single search terms paired with the name of each African country, or were search term combinations paired with the name of each African county? In which search contexts were each of the "OR" and "AND" Boolean operators used? 	Thank you very much for drawing our attention to this. We have amended same to clarify the process in lines 96-105
 There are a lot of acronyms in this paper (for example, see lines 91-92); adding an abbreviations section before the introduction would help readers keep track of these. 	Thanks for drawing our attention to these. The acronyms we used are standard abbreviations used across clinical microbiology, hence we did not expect it to be a problem. However, if the Editor/journal will accept this as part of their formatting, we shall then add an abbreviation section above the introduction. Notwithstanding, we tried providing the meaning of all

	abbreviations in-text before using them subsequently as is standard practice.
 • The authors use the term 'frequency' to describe their data (e.g. line 109), but the data presented here are actually counts. As far as I can tell, the numbers presented in the supplemental tables and figures are all counts. Frequencies would represent some proportion of a dataset, rather than a raw number. 	Thank you sincerely for drawing our attention to these. Indeed, we are not very versed in statistics, so we apologise for this blunder. We have amended same throughout the text.
 • The supplemental information contains an impressive amount of information, but is difficult to read. I would recommend clarifying in the text what is represented in each of the supplemental files (e.g. Table S1 is GNBs isolated from animal sources, Table S2 is GNBs isolated from human sources). I also think the inclusion of the graphs in these supplemental data files is confusing and does not add further analysis or information; it is just another way of displaying the same data. I would recommend removing the graphs entirely and keeping the tables. As the information in these figures is already present in the supplemental tables, I would also recommend removing S1-S6. 	We agree and have removed figures S1-S6 as well as several figures in Tables S1-S3, leaving a skeletal few for pictorial purposes for those who can go through the Tables. The content of each supplemental file is also described in lines 129, 160 & 163.
 • The figures in this paper need work. The country names in Supplemental figures S1 and S6 are upside down and all the figure panels have been stretched to be almost unreadable. 	The figures S1-S6 have been removed. High-quality production images are large and cannot be uploaded, but will be made available during production.
 • This manuscript is missing any statistical analyses of the data. I think the study would be greatly strengthened by the addition of statistical analyses such as the following: 	Thank you very much for your insights and help. Although this study was not designed as a meta-analyses paper, we have tried to insert statistical component using Pearson's Chi-square test without success as using your calculations provided us with non-integer expected values. Whiles we are most grateful for this direction, we do not currently have the capacity to make this a meta-analyses. We have however provided the calculations as directed in Table S7 for readers to see the relative ratios of the various ARGs, MGEs, Species, and resistomes per countries and ecological niche..
 o A statistical comparison of the species distributions identified in different countries (section starting on line 154). First, the proportion of isolates associated with each species should be calculated as (number of isolates of Species X / total number of isolates identified from that country). Then, those proportions should be statistically compared between countries. This could be done by pairwise testing (with multiple-test corrections) or by a chi-square test. 	
 o A similar statistical comparison between the proportion of isolates of a given species associated with human, animal, or environmental samples within a country. 	
 o The same analyses should be completed for clones, ARGs, and MGEs. For the ARGs, differing mechanisms of resistance to the same antibiotic could be grouped together for increased statistical power. 	
 o The richness of the resistome in a given species should be statistically compared as well. In the discussion, the authors mention that E. coli does not have the richest resistome, but this seems to be based on counts alone. It would be interesting to statistically compare the number/richness of resistance genes in each species to strengthen this argument. 	
 • For the section on resistance rates (line 285), I have the following comments: 	Thanks. Rates have been changed to levels throughout the text.
 o I think "resistance levels" is a more appropriate term than "resistance rates": rate implies a time component. 	
 o There is no explanation about how resistance levels were compared between different resistance metrics (e.g. E-test vs. microdilution); different tests of antibiotic resistance are known to give very different answers as to level of resistance. 	In calculating the resistance levels of each species to an antibiotic per country, we used absolute cut-offs of either resistant or susceptible i.e., the isolate was defined as either resistant
 o The way the manuscript and Tables S1-S3 are currently laid out, 	

it seems like the authors consider antibiotic resistance a binary trait; that is, a strain/species is either resistant to a given antibiotic, or it is not. This is an unusual way to define antibiotic resistance levels, which are usually reported as a quantitative value (minimum inhibitory concentration MIC, or minimum bacteriocidal concentration, MBC). I am not sure how the papers included in this meta-analysis reported antibiotic resistance, but if they reported a value such as MIC or MBC, the authors of this paper should include a table of cutoff values wherein they delineate 'resistant' vs. 'non-resistant' strains.	or susceptible to the antibiotic in question. The type of diagnostic or breakpoint benchmark viz., CLSI or EUCAST, used to measure define the AR in the isolates were not considered in this reckoning. Thus, if the authors defined the isolate as resistant, it was used as such in calculating the resistance levels. Not all the included studies reported the AR state of their isolates using the MIC. Some used disc diffusion. To avoid complicating the analyses, we just defined the isolates as resistant or susceptible and this is not unusual; this is acceptable standards by the the CLSI/EUCAST and is what is used in clinical microbiology labs. See lines 129-137
 Line 391-393: I am very interested in this apparent outbreak of clonal Salmonella enterica serovar Typhi that the authors uncovered. Is there a way of incorporating a time-series analysis to associate, for example, sample collection date with the location of this strain? This information might then be used to trace the source of the outbreak. I think this would be a very interesting finding from these data, and greatly strengthen the paper. 	We appreciate the reviewer's interest in this. We do not have skill to undertake this time-series analyses, unless the reviewer can explain it further to us so we can do it. However, we refer him/her to the new figures 9-11, which shows the bootstrap values of each branch as well as the year of isolation of each isolate and the country of isolation in the labels. We believe that the year (time) component and the geographical location of the isolates in the label, coupled with the evolutionary relationship between the isolates in the figures 9-11 answers to this interest. We invite the reviewer to enlarge this newer and better image to appreciate this. Please see lines 431-432
Minor comments:	
 Antibiotic resistance is abbreviated as "AR" and "ABR" in different locations in the paper. 	Thanks. All have been changed to AR throughout the text
 Line 24-26: minor grammatical comments for this sentence: "...ramifications in developing countries ARE worsened by..." and "inadequate NUMBERS OF skilled clinicians and scientists..." 	Thanks. Corrections effected.
 Line 45: the last sentence of this paragraph seems out of place. Adding something at the beginning such as "These results clearly indicate that One Health studies in Africa are needed." 	Thanks. Sentence has been removed.
 Line 55: grammar edit: instead of "Coupled with these", consider "In addition to these issues, there is limited funding..." 	Sentence corrected. In order to stay within the word limit of the journal, we try to shorten sentences with words that carry the same meaning with little explanation.
 Line 58: the word "Animals" should be "animals" (no 	Thanks. Corrected.

capitalization).	
• Line 66: consider saying "...ARGs are found in more diverse GNB species and genera, confer multi-drug resistance, and are associated with substantial morbidities and mortality."	Thanks. Sentence has been modified as suggested in lines 68-69 .
• Line 112: how did the authors decide which genera should have their genomes downloaded from PATRIC.	Genomes of GNB, which were not reported in the included articles and that had no genomes on PATRIC were not included. Lines 142-155
• Line 118-119: what were the search parameters used for the Isolates Browser?	The genomes' accession or biosample number was used to retrieve their ARGs. See lines 125-126
• Line 125: what is meant by "manual search"?	This has been revised. They were obtained from the references of the included articles.
• Line 126-127: what was the screening process used to get from 868 to 309 manuscripts? From 309-146? I assume this screening was based on the inclusion/exclusion criteria described in the methods, but that needs to be explicitly stated here.	Thanks. These have been stated in lines 109--115
• Line 129-140: These data should be displayed in a table or graph (preferably the latter).	Thanks for the suggestion. This has been done as Fig. 1B. lines 174-177
• Table 2 should go in the supplemental information.	This has been moved to Table S7
• Line 149" CAR and DRC need to be written out.	Corrected in line 189
• Line 153: no need to reference the tables/figures here; presumably these countries are reflected in most of the paper's figures and tables.	Table and figure referencing removed.
• Line 156,334: "most common" rather than "commonest"	Changed throughout the manuscript
• Line 162: how are the authors determining which species are most concentrated where?	These were determined based on the counts of each pathogen per country: L205
• Line 214-228: There is a lot of data presented in the text (n=___). The authors should instead reference the table where these counts are found (once).	Thank you. These have been amended as suggested.
• Line 261-262: A/H/E need to be defined before they are used. Adding a sentence such as "A= animal source, H= human source, E= environmental source" would be helpful here.	These have been previously defined under results in line 138, need we define them again? We defined again in line 299
• Line 347: I would be interested in what is known in the literature about ARG and GNB distribution on other continents as a comparison to the findings of this paper. Have similar meta-analyses been done for different continents that could be discussed in this part of the paper?	We did not find a similar meta-analysis or phylogenetics studies or calculations on the subject in other continents. This made comparison impossible.
• When discussing Salmonella strains, the correct nomenclature is S. enterica serovar Typhi (e.g. in line 383).	S. enterica was used here because there were several serovars besides Typhi and stating all the serovars would be too much.

October 15, 2020

Dr. John Osei Sekyere
University of Pretoria
Department of Medical Microbiology
School of Medicine
Faculty of Health Sciences
Pretoria, Gauteng 0084
South Africa

Re: mSystems00897-20 (Genomic and Resistance Epidemiology of Gram-Negative Bacteria in Africa: A Systematic Review and Phylogenomic analyses from a One-Health Perspective)

Dear Dr. John Osei Sekyere:

The reviewers and I have found that you have addressed many of the comments, and that this manuscript will be impactful once published. However, there are a few additional comments that should be addressed before publication.

Below you will find the comments of the reviewers.

To submit your modified manuscript, log onto the eJP submission site at <https://msystems.msubmit.net/cgi-bin/main.plex>. If you cannot remember your password, click the "Can't remember your password?" link and follow the instructions on the screen. Go to Author Tasks and click the appropriate manuscript title to begin the resubmission process. The information that you entered when you first submitted the paper will be displayed. Please update the information as necessary. Provide (1) point-by-point responses to the issues raised by the reviewers as file type "Response to Reviewers," not in your cover letter, and (2) a PDF file that indicates the changes from the original submission (by highlighting or underlining the changes) as file type "Marked Up Manuscript - For Review Only."

Due to the SARS-CoV-2 pandemic, our typical 60 day deadline for revisions will not be applied. I hope that you will be able to submit a revised manuscript soon, but want to reassure you that the journal will be flexible in terms of timing, particularly if experimental revisions are needed. When you are ready to resubmit, please know that our staff and Editors are working remotely and handling submissions without delay. If you do not wish to modify the manuscript and prefer to submit it to another journal, please notify me of your decision immediately so that the manuscript may be formally withdrawn from consideration by mSystems.

Sincerely,

Zarath Summers

Editor, mSystems

Journals Department
Reviewer comments:

Reviewer #1 (Comments for the Author):

The authors have improved their manuscript significantly. However, there are still some issues from my initial review that they have not addressed; I encourage them to do so.

Very minor comments:

Please state the parameters for running RAxML phylogeny apart from the 1000 bootstrap analysis. Which model was used to run the ML phylogeny?

I did not find the explanation for the *A. baumannii* MLST in Table S4.

For many of the phylogenetic trees the branches are still transformed. When visualizing the tree in FigTree, you want to unselect the "Transform branches" option under the Trees panel. As the trees stand right now, a reader cannot infer the real genetic distance among the taxa in those particular trees.

Reviewer #3 (Comments for the Author):

The goal of this study was to investigate the geographical and phylogenetic distribution of antibiotic resistance (AR) genes in several African countries. To this end, the authors conducted a broad meta-analysis of English-language articles on bacterial infections in 41 countries published between 2015-2019. The authors identified several Gram-negative pathogens as frequent sources of AR genes, and recommended using *Enterobacter* spp. Or *Klebsiella pneumoniae* as sentinel species for AR surveillance in the surveyed areas. This is a novel, well executed study that fits well within the scope of mSystems. However, there are a few issues, as described below.

Overall, the study is interesting, important, and novel. The authors seem to have integrated previous reviewers' comments very well into this manuscript. The dataset collected for this study is impressive in its breadth and depth, and represents a rich source of information for AR surveillance studies in Africa. The manuscript is well written and clear. My most significant concern is the presentation of data in the figures and tables. All of the supplemental tables contain raw data as well as analysis results, making the take-home message of each table difficult to discern. There are also far too many main figures, mostly showing the same analyses for different pathogens. My two overall recommendations are to remove raw data from the supplemental tables and condense them into smaller summaries (averages, ratios, etc. as appropriate), and to move most of the main figures to the supplement. Further comments are described in more detail below.

Major comments:

- There is an impressive amount of data and figures for this paper- far too many figures to include in the main text. I recommend that the authors choose 3-5 pathogens of particular interest to include in the main body of the text, and move the rest of figures 2-26 to the supplemental information. Perhaps the genera that are most frequently observed in human samples? (*Escherichia*, *Salmonella*, *Klebsiella*, *Pseudomonas*, and *Acinetobacter*)? Or the genera that are found in the greatest number of countries?
- Line 332 and following section: I would recommend caution with the language stating that a strain "expressed very high resistance". From the methods, it seems like what is being surveyed is a binary measure of resistance based on EUCAST or CLSI. In this case, a given isolate/species doesn't have a "level of resistance" associated with it; it's either resistant or not. It seems like what the authors are trying to say is that the number or frequency of genes conferring resistance to various antibiotics are highest among human strains (in line 333-334, for example). The language in this section should be changed from high/low resistance levels to high/low prevalence of resistance genes.
- Many of the figures in this paper are examining a particular species/genus. This is interesting, but also makes it difficult to compare results across different genera. Particularly since the authors mention early in the discussion that *Enterobacter* might make a better surveillance organism than *E. coli* due to its richer AR gene repertoire, I would be interested in seeing an additional figure illustrating this. Perhaps as an additional panel in figure 1, similar to figure 1B but with number of unique AR genes identified (or number of antibiotics for which resistance genes were identified) in each country.
- The supplemental tables essentially represent the raw data that the authors have collected, rather than a summary of important features of that data. I would recommend that the authors choose one supplemental file to contain all their raw data (in multiple tabs, if necessary) and remove any other supplemental tables. Alternatively, the authors could add a line in the manuscript indicating that data are available upon request.
- One future extension I would recommend mentioning is a statistical analysis of these data. The qualitative and count results presented here are interesting and useful, and will hopefully be a starting point for deeper quantitative analysis of this impressive dataset.

Minor comments:

- Line 10-12: I think a tweet about this article is an excellent idea. I recommend strengthening the language to emphasize the novelty and importance of this work. For example: Antibiotic resistance is an understudied public health threat in Africa. Here, we provide the first comprehensive analysis of AR gene epidemiology and phylogenomics in Gram-negative pathogens from a One Health perspective.
- Line 31: PATRIC should be defined, as should RAXML; these tools are commonly used in microbiology, but I think this paper will have a broad audience and should therefore have acronyms

defined as much as possible.

- Line 73: I would like to see a specific definition of One Health surveillance. I think this would greatly help in illustrating why this study is novel and important.
- Line 76-86: recommend clarifying this sentence: "The importance of this One Health conflict lies in the sources of AR. When AR-containing bacteria from farms end up in the environment through effluents and manure application on soils, they can be transferred to humans through vegetable and animal diets."
- Line 152: Streptococcus should be capitalized.

mSYSTEMS	
Reviewer #1 (Comments for the Author):	AUTHOR RESPONSES
The authors have improved their manuscript significantly. However, there are still some issues from my initial review that they have not addressed; I encourage them to do so.	Thanks for the compliments. The issues are described below.
Very minor comments:	
Please state the parameters for running RAxML phylogeny apart from the 1000 bootstrap analysis. Which model was used to run the ML phylogeny?	Version, other parameters, and model used are stated in lines 160-163
I did not find the explanation for the A. baumannii MLST in Table S4.	Please check row 24 in Table S4 and lines 976-978 under Table S4's legend in the main text.
For many of the phylogenetic trees the branches are still transformed. When visualizing the tree in FigTree, you want to unselect the "Transform branches" option under the Trees panel. As the trees stand right now, a reader cannot infer the real genetic distance among the taxa in those particular trees.	Indeed, we do agree with the reviewer that some of the trees were not transformed. This is not a deliberate action on our part. However, for these trees, which are in the minority compared to the majority that could be transformed, the conformation and text were so changed that they could not be legible when transformed. Thus, for clarity reasons, we decided to maintain it in a transformed state to enhance the aesthetics. We are particularly sorry for this, but as it stands, it is beyond our control for now. Some of these figures are now supplementary.
Reviewer #3 (Comments for the Author):	
The goal of this study was to investigate the geographical and phylogenetic distribution of antibiotic resistance (AR) genes in several African countries. To this end, the authors conducted a broad meta-analysis of English-language articles on bacterial infections in 41 countries published between 2015-2019. The authors identified several Gram-negative pathogens as frequent sources of AR genes, and recommended using Enterobacter spp. Or Klebsiella pneumoniae as sentinel species for AR surveillance in the surveyed areas. This is a novel, well executed study that fits well within the scope of mSystems. However, there are a few issues, as described below.	Thank you for your time...the issues are addressed below.
Overall, the study is interesting, important, and novel. The authors seem to have integrated previous reviewers' comments very well into this manuscript. The dataset collected for this study is impressive in its breadth and depth, and represents a rich source of information for AR	We agree with the reviewer and appreciate his concerns. We have stipulated our specific responses below. We tried moving some of the figures to

surveillance studies in Africa. The manuscript is well written and clear. My most significant concern is the presentation of data in the figures and tables. All of the supplemental tables contain raw data as well as analysis results, making the take-home message of each table difficult to discern. There are also far too many main figures, mostly showing the same analyses for different pathogens. My two overall recommendations are to remove raw data from the supplemental tables and condense them into smaller summaries (averages, ratios, etc. as appropriate), and to move most of the main figures to the supplement. Further comments are described in more detail below.	supplemental figures, but this infringed on mSystems editorial requirements of not having more than 10 supplemental files or having a single PDF of all supplemental figures. So, we were forced to revert back to putting all the images in the main figures text. Fortunately, mSystems does not restrict us on the number of figures. Table S7 presents a summary of all the data as suggested by the reviewer. The other supplementary Tables represent a very dicey situation due to the nature of the study and enormity of the data from (1) literature and (2) genomics. First, the Tables have been separated into animals, humans, and environment with their associated analyses in separate tabs. Second, the genomic aspect of the study is also separated into their distinct Tables with their associated analyses.
Major comments:	
 • There is an impressive amount of data and figures for this paper- far too many figures to include in the main text. I recommend that the authors choose 3-5 pathogens of particular interest to include in the main body of the text, and move the rest of figures 2-26 to the supplemental information. Perhaps the genera that are most frequently observed in human samples? (Escherichia, Salmonella, Klebsiella, Pseudomonas, and Acinetobacter)? Or the genera that are found in the greatest number of countries? 	Thanks. We tried moving some of the figures to supplemental figures, but this infringed on mSystems editorial requirements of not having more than 10 supplemental files or having a single PDF of all supplemental figures. So, we were forced to revert back to putting all the images in the main figures text. Fortunately, mSystems does not restrict us on the number of figures.
 • Line 332 and following section: I would recommend caution with the language stating that a strain "expressed very high resistance". From the methods, it seems like what is being surveyed is a binary measure of resistance based on EUCAST or CLSI. In this case, a given isolate/species doesn't have a "level of resistance" associated with it; it's either resistant or not. It seems like what the authors are trying to say is that the number or frequency of genes conferring resistance to various antibiotics are highest among human strains (in line 333-334, for example). The language in this section should be changed from high/low resistance levels to high/low prevalence of resistance genes. 	The reviewer is right. We have thus changed this wording to resistance rate to reflect percentage resistance among the isolates. The data being discussed in this section is not the ARGs, but the phenotypic resistance measured in the various articles. Hence, ARGs cannot be used here. Therefore, our decision to use resistance rates to represent percentage of resistance among the isolates.
 • Many of the figures in this paper are examining a particular species/genus. This is interesting, but also makes it difficult to compare results across different genera. Particularly since the authors mention early in the 	This data is already in supplemental Table S7...However, we have also added it to Fig 1. as 1C. See lines 852-853.

discussion that Enterobacter might make a better surveillance organism than E. coli due to its richer AR gene repertoire, I would be interested in seeing an additional figure illustrating this. Perhaps as an additional panel in figure 1, similar to figure 1B but with number of unique AR genes identified (or number of antibiotics for which resistance genes were identified) in each country.	
 The supplemental tables essentially represent the raw data that the authors have collected, rather than a summary of important features of that data. I would recommend that the authors choose one supplemental file to contain all their raw data (in multiple tabs, if necessary) and remove any other supplemental tables. Alternatively, the authors could add a line in the manuscript indicating that data are available upon request. 	Table S7 presents a summary of all the data as suggested by the reviewer. The other supplementary Tables represent a very dicey situation due to the nature of the study and enormity of the data from (1) literature and (2) genomics. First, the Tables have been separated into animals, humans, and environment with their associated analyses in separate tabs. Second, the genomic aspect of the study is also separated into their distinct Tables with their associated analyses. Combining them into a single file will rather increase the confusion for the reader. mSystems also does not allow authors to combine separate supplemental Tables into a single excel file or to restrict data; all data are to be made publicly available either as supplemental files or in a public repository. Hence, I can not implement this without going against their editorial requirements.
 One future extension I would recommend mentioning is a statistical analysis of these data. The qualitative and count results presented here are interesting and useful, and will hopefully be a starting point for deeper quantitative analysis of this impressive dataset. 	Thanks. This has been included in lines 591-594
Minor comments:	
 Line 10-12: I think a tweet about this article is an excellent idea. I recommend strengthening the language to emphasize the novelty and importance of this work. For example: Antibiotic resistance is an understudied public health threat in Africa. Here, we provide the first comprehensive analysis of AR gene epidemiology and phylogenomics in Gram-negative pathogens from a One Health perspective. 	Thanks. This suggested change has been effected.
 Line 31: PATRIC should be defined, as should RAXML; these tools are commonly used in microbiology, but I think this paper will have a broad audience and should therefore have acronyms defined as much as possible. 	Thanks. This has been done.
 Line 73: I would like to see a specific definition of One Health surveillance. I think this would greatly help in 	Please see line 76-77 for this definition.

illustrating why this study is novel and important.	
 Line 76-86: recommend clarifying this sentence: "The importance of this One Health conflict lies in the sources of AR. When AR-containing bacteria from farms end up in the environment through effluents and manure application on soils, they can be transferred to humans through vegetable and animal diets." 	This has been modified: lines 77-80
 Line 152: Streptococcus should be capitalized. 	Done: line 155

October 28, 2020

Dr. John Osei Sekyere
University of Pretoria
Department of Medical Microbiology
School of Medicine
Faculty of Health Sciences
Pretoria, Gauteng 0084
South Africa

Re: mSystems00897-20R1 (Genomic and Resistance Epidemiology of Gram-Negative Bacteria in Africa: A Systematic Review and Phylogenomic analyses from a One-Health Perspective)

Dear Dr. John Osei Sekyere:

Your manuscript has been accepted, and I am forwarding it to the ASM Journals Department for publication. For your reference, ASM Journals' address is given below. Before it can be scheduled for publication, your manuscript will be checked by the mSystems senior production editor, Ellie Ghatineh, to make sure that all elements meet the technical requirements for publication. She will contact you if anything needs to be revised before copyediting and production can begin. Otherwise, you will be notified when your proofs are ready to be viewed.

Sincerely,

Zarath Summers
Editor, mSystems

Journals Department
Table S5: Accept
Table S2: Accept
Figure S3: Accept
Figure S2: Accept
Table S6: Accept
Table S1: Accept
Table S3: Accept
Table S4: Accept
Table S7: Accept
Figure S1: Accept